# EnsemDeepCADx: Empowering Colorectal Cancer Diagnosis with Mixed-Dataset Features and Ensemble Fusion CNNs on Evidence-Based CKHK-22 Dataset

**DOI:** 10.3390/bioengineering10060738

**Published:** 2023-06-19

**Authors:** Akella Subrahmanya Narasimha Raju, Kaliyamurthy Venkatesh

**Affiliations:** Department of Networking and Communications, School of Computing, SRM Institute of Science and Technology, SRM Nagar, Chennai 603203, India; venkatek2@srmist.edu.in

**Keywords:** colorectal cancer, ensemble fusion CNNs, bidirectional long short-term memory (BiLSTM), support vector machine (SVM), CKHK-22 mixed dataset, feature fusion

## Abstract

Colorectal cancer is associated with a high mortality rate and significant patient risk. Images obtained during a colonoscopy are used to make a diagnosis, highlighting the importance of timely diagnosis and treatment. Using techniques of deep learning could enhance the diagnostic accuracy of existing systems. Using the most advanced deep learning techniques, a brand-new EnsemDeepCADx system for accurate colorectal cancer diagnosis has been developed. The optimal accuracy is achieved by combining Convolutional Neural Networks (CNNs) with transfer learning via bidirectional long short-term memory (BILSTM) and support vector machines (SVM). Four pre-trained CNN models comprise the ADaDR-22, ADaR-22, and DaRD-22 ensemble CNNs: AlexNet, DarkNet-19, DenseNet-201, and ResNet-50. In each of its stages, the CADx system is thoroughly evaluated. From the CKHK-22 mixed dataset, colour, greyscale, and local binary pattern (LBP) image datasets and features are utilised. In the second stage, the returned features are compared to a new feature fusion dataset using three distinct CNN ensembles. Next, they incorporate ensemble CNNs with SVM-based transfer learning by comparing raw features to feature fusion datasets. In the final stage of transfer learning, BILSTM and SVM are combined with a CNN ensemble. The testing accuracy for the ensemble fusion CNN DarD-22 using BILSTM and SVM on the original, grey, LBP, and feature fusion datasets was optimal (95.96%, 88.79%, 73.54%, and 97.89%). Comparing the outputs of all four feature datasets with those of the three ensemble CNNs at each stage enables the EnsemDeepCADx system to attain its highest level of accuracy.

## 1. Introduction

Global health has a wide range of complicated aspects. The development of globalisation has had a significant negative impact on people’s health across the world. People who reside in underdeveloped countries, where such services may be rare, may be significantly affected by environmental factors such as air pollution, contaminated water, and poor sanitation [1]. Despite having improved access to medical care, people living in wealthy countries still confront challenges, including increased incidence of non-communicable diseases such as obesity, diabetes, and heart disease brought on by sedentary lives and bad eating practices. Cultural differences have a big influence on health outcomes, too. The great variety of cultural practices that exist may make it more difficult to promote universal health policies and strategies. All individuals must have access to healthcare, education, and resources to promote healthy lifestyle habits. Environmental problems that contribute to health inequities must also be addressed.

Cancer, one of the leading causes of mortality, affects individuals all over the globe. It is a complex problem that requires a comprehensive approach to diagnosis, care, and prevention. According to the World Health Organization (WHO), there will be approximately 19 million new cases of cancer worldwide in 2020. In 2020, cancer will be responsible for an estimated 9.9 million fatalities worldwide [2]. Seventy percent of all cancer fatalities occur in low- and middle-income countries, indicating that the disease burden is not distributed evenly across the globe. Access disparities to cancer-trained healthcare professionals exacerbate the difficulty of coping with the disease. According to the World Health Organization, there is a global scarcity of 18 million healthcare personnel, with the shortage being most severe in countries with low per capita incomes. It is estimated that there are 1.5 physicians per 1000 persons in the world, but this figure varies widely based on location and level of prosperity [3]. In low- and middle-income countries, the lack of qualified medical personnel is a significant barrier to providing effective cancer treatment. The distribution of healthcare, particularly cancer services, must be enhanced across regions and socioeconomic classes, necessitating the development of policies to increase the number of healthcare workers. Additionally, there is a need for increased investment in research in order to improve cancer care delivery worldwide and develop more effective treatments for the disease.

An estimated 1.9 million new instances of colorectal cancer (which may affect either the colon or the rectum) and 935,000 deaths were recorded worldwide in 2020, according to the World Health Organization [4]. It is the second leading cause of cancer mortality and the third most prevalent malignancy diagnosed globally. Colorectal cancer is increasingly prevalent in wealthy nations, although its occurrence varies widely around the globe. According to the American Cancer Society, the United States has the highest prevalence of colorectal cancer in the world, with an estimated 145,600 new cases and 51,000 deaths in 2019 [5]. It is estimated that there are around 45,000 new instances of colorectal cancer diagnosed each year in India, making it the fourth most frequent disease there. Colorectal cancer rates have been increasing at a rate of 1.8% per year in India between 1990 and 2016. Cities such as Delhi, Mumbai, and Kolkata have a greater prevalence of colorectal cancer than rural parts of India [6]. The rising trend in colorectal cancer incidence makes it a major health issue even if the incidence is lower in India than in industrialised nations. Particularly in outlying places, it is difficult to have access to adequate screening programs and medical treatment [7]. Raising awareness and funding programs for early diagnosis and treatment are essential steps towards reducing the prevalence of colorectal cancer in India.

A multitude of variables, including but not limited to age, family history, and lifestyle factors such as smoking, alcohol use, lack of physical activity, and poor nutrition, may contribute to the high incidence of colorectal cancer. Late diagnosis and poor outcomes are worsened in low- and middle-income countries by a lack of access to quality screening programs and healthcare services. Despite these challenges, colorectal cancer may be avoided if it is discovered early and treated immediately. Screening procedures such as colonoscopy, stool-based testing, and computed tomography colonography (CT) may all detect precancerous or early-stage cancers, allowing for more successful treatment [8].

Colonoscopy is the most reliable method for detecting and diagnosing colorectal cancer at an early stage. The rectum and colon are inspected for polyps and other growths using a flexible conduit equipped with a miniature camera [9]. Colonoscopy has several advantages over other colorectal cancer screening methods, including the capacity to detect and remove polyps. To stop the cancer from spreading to other regions of the body, the malignant cells must be removed or eradicated. Cancer survival and recurrence rates could be substantially improved by early detection and treatment. Screenings for colorectal cancer should be routine. Those who are at risk for colorectal cancer due to their age or other factors should receive regular screenings. In the United States and Canada, those at ordinary risk should commence colorectal cancer screening at age 45. Beginning at age 50, screening is advised in the United Kingdom and other countries. Those with a personal or familial history of colorectal cancer, as well as those with additional risk factors, may be advised to begin screening earlier or more frequently.

Images collected during colonoscopy, a common colorectal cancer screening method, can be used to train artificial intelligence (AI), machine learning (ML), and deep learning (DL) systems for computer-aided diagnosis (CADx) [10]. By analysing and interpreting the colonoscopy images, these diagnostic tools may facilitate a more precise diagnosis of colorectal cancer. Artificial intelligence (AI), machine learning (ML), and deep learning (DL) enable CADx systems to process enormous amounts of data quickly and provide physicians with instantaneous results. Consequently, patients with colorectal cancer may receive treatment more quickly and experience improved outcomes. In collaboration with the advancement of AI and ML technologies, the potential for CADx systems to considerably enhance colorectal cancer screening and detection is growing.

Convolutional Neural Networks (CNNs) are an effective method for detecting and diagnosing colorectal cancer when applied to colonoscopy images. Convolutional neural networks (CNNs) are well-suited for identifying polyps and other aberrant growths in the colon because they evaluate visual input and discover patterns using deep learning techniques. In recent years, the use of ensemble learning approaches to improve CNNs’ ability to detect colorectal cancer has been investigated [11,12]. Fusion CNNs is one such technique; it integrates multiple CNN models to create a more dependable and accurate system. Researchers have demonstrated that fusion CNNs are more effective at diagnosing colorectal cancer than standalone CNN models. Fusion CNNs combine the strengths of multiple CNN models to compensate for their shortcomings and provide more precise and reliable diagnoses [13]. The advancement of these technologies bodes well for their application in the early detection and diagnosis of digestive tract malignancies.

Diverse methods, such as Fusion CNNs, bidirectional long short-term memory (BILSTM) networks, and support vector machines (SVMs), have been used to examine the accuracy of identifying colorectal cancer from colonoscopy images. The term “transfer learning” refers to the process of adapting a neural network’s training on one task to another task that is similar [14]. A neural network can be trained on a massive dataset of generic medical images using transfer learning in order to detect colorectal cancer. BILSTMs are a type of recurrent neural network that can process data sequences both forward and backward. BILSTMs have been demonstrated to be effective for analysing colonoscopy images due to their ability to detect both local and global patterns. Support vector machines (SVMs) are classification applications of supervised learning algorithms. They function by identifying the optimal hyper-plane for data clustering. There is evidence that combining SVMs with BILSTMs and transfer learning improves the diagnostic accuracy of colorectal cancer.

The proposed CADx for colorectal cancer detection, the EnsemDeepCADx system, employs a multistage technique in order to locate and identify polyps that may be indicative of the disease.

The initial stage is to generate a dataset for feature fusion by combining the CKHK-22 mixed image datasets with others, such as the Grey and LBP datasets.Developing and training three ensemble fusion CNNs using the feature fusion CNN and the other featured datasets is the next step.In the subsequent stage, all CNNs are trained with SVM and transfer learning.In the final stage, temporal and spatial information is extracted using transfer learning ensemble fusion CNNs with BILSTM and SVM multi-class classification.At each stage, the EnsemDeepCADx model is inspected for inconsistencies and other performance metrics, and the resulting data is compared to determine the optimal approach.

### 1.1. Organisation of Study

The research article is divided into multiple sections. The second section provides a comprehensive literature review of the extant studies and research relevant to the proposed CADx system for diagnosing colorectal cancer. In Section 3, the study’s materials and methods are enumerated and explained in depth. This comprises the dataset used, methodologies for feature extraction, and training and testing procedures for the proposed CADx system. The results of the experiments, including performance metrics, are presented and discussed in Section 4. Section 5 concludes the research by summarising the main findings, discussing the study’s limitations, and proposing future research directions in this area.

### 1.2. Literature Survey

Alba Nogueira-Rodriguez et al. [15], who identified and categorised polyps using DL techniques, extensively discussed polyps as significant precursors to colorectal cancer. As inputs for DL-based systems, they examined colonoscopy datasets for public and commercial use. In addition, the authors examined the numerous DL technologies currently used in cancer diagnosis. Specifically, they investigated the DL-based CADx system used to detect malignancy on computers. Precision, recall accuracy, and F1 score are all valuable metrics for evaluating the classification performance of a DL-based system. The authors also discussed the difficulty of accurately identifying lesions smaller than 5 mm and how DL techniques may be able to assist with this issue. Although this study employs DL to identify cancer lesions, the authors do not provide a streamlined method.

Hwang, Y.K. et al. [16] proposed a method that employs convolutional neural networks (CNNs) to automatically identify and classify colorectal lesions. This study utilised a compilation of colonoscopy images obtained from the Tada Tomohiro Institute of Gastroenterology and Proctology in Japan. The compilation included 16,418 images from 12,895 patients between December 2013 and March 2017. After being pre-processed, the enhanced images were used in the system’s training and testing phases. For polyp detection and classification, the system utilised the Single Shot Multibox Detector (SSD), a neural network with 16 convolutional layers. Caffe, a framework for deep learning, was used to construct and test CNN. White-light imaging facilitated an 83% detection rate for colorectal lesions and a 97% detection rate for adenomas. However, pre-trained or isolated datasets were not used to evaluate the effectiveness of the system in this study.

Omneya Attallah et al. [17] introduced gastro-CADx for the first time, which is a deep learning-based method for categorising numerous GI disorders. This technique is composed of three stages. Initially, spatial information is extracted using four distinct CNNs as feature extractors. In the second phase, the temporal-frequency and spatial-frequency characteristics are used as inputs for the discrete wave transform (DWT) and discrete cosine transform (DCT), respectively. In the final phase, the optimal feature set is determined by combining multiple feature sets and analysing their influence on the CADx output results. Dataset I is the Kvasir dataset, while Dataset II is the Hyper Kvasir dataset. The concept of semantic segmentation for the localisation and detection of malignant polyps has not been clarified, nor has the method been validated with additional datasets.

Muthu Subhash Kavitha et al. [18] highlighted convolutional neural networks (CNN) as a potential solution for datasets consisting of colonoscopy images in their discussion of deep learning approaches for the early diagnosis of colorectal cancer. The researchers compared numerous techniques for identifying and localising lesions in the colorectal region, such as CNN transfer learning, end-to-end learning, hybrid learning, and explainable AI. However, the authors discussed only the theoretical aspects of these methodologies and made no reference to experimental or actual work.

Zheng Cao et al. [19] proposed using Raman spectra and a method of deep learning to identify colorectal cancer cases. The authors obtained Raman spectroscopy data from 26 patients with colorectal cancer. The Raman displacements of the patients ranged from 385 cm^−1^ to 1545 cm^−1^. These datasets were submitted to a 1D-ResNet CNN to generate classifications. The investigation revealed a detection success rate of 98.5% for colorectal cancer. One CNN model was employed for cancer identification, but it was not evaluated on other image datasets. It is also believed that Raman spectroscopy of the colorectal region is a time-consuming and laborious process.

Mahmoud Ragab et al. [14] provided the optimal deep transfer learning strategy for the early detection of colorectal cancer. This study detected colorectal cancer using an algorithm inspired by slime mould (SMADTL-CCDC) and deep transfer learning. The primary objective of the SMADTL-CCDC strategy is to detect colorectal cancer at an early stage. This study employed the dense-EfficientNet method to generate feature vectors from the pre-processed images. We were able to identify and classify cases of colorectal cancer using SMADTL-CCDC and the Discrete Hopfield Neural Network (DHNN) method. The SMADTL-CDC model performed better than more recent methods. For testing, fewer histopathological datasets were utilised than for training; the ratio was 90:10. Consequently, the concept’s utility may be limited.

Saban Ozturik et al. [20] investigated the efficacy of artificial intelligence methods in producing an accurate colon cancer diagnosis. In particular, the study compared the efficacy of numerous CNN models on a small collection (2000–6000 images) of colonoscopy image datasets. Each CNN pooling layer from models such as AlexNet, GoogleNet, and ResNet50 was converted into an LSTM layer for use in the overall classification procedure. There were only three CNN models examined, and a large reference dataset for colonoscopy was not utilised.

Using a deep learning-based polyp detector, Meryem Souaidi et al. [21] hypothesised an anomaly in the polyp area of WCE and colonoscopy localisation and visualisation. The authors recommend an MP-FSSD polyp identification model constructed on VGG-16 backbones. However, only convolutional neural network (CNN) models such as VGG-16 are employed, and neither their benefits nor drawbacks are discussed in this study. In addition, the authors analysed only the WCE and CVC clinic database sets for polyp detection.

Nguyen Thanh Duc et al. [22] proposed a novel deep learning technique for detecting lesions in the colon. To aid in the detection of lesions in colonoscopy images, scientists have developed “ColonFormer”, a deep learning architecture based on the concept of semantic segmentation. In the proposed encoder–decoder architecture, a lightweight encoder modelling global semantic connections across scales is combined with a hierarchical decoder to enhance feature representations. Using five distinct reference datasets, the authors evaluated the efficacy of the proposed method. This model incorporates the advantages of a transformer and a CNN to generate accurate multiscale representations of features. It can, however, only analyse data from five distinct datasets and a single architecture. The findings of the breakthrough comparison analysis strongly suggest that this strategy has produced the best results.

Saito et al. [23] used anatomical images from robotic colonoscopies in a study that relied on classification by a deep convolutional neural network. The most prevalent locations for colon cancer are the terminal ileum, cecum, ascending colon, transverse colon, descending colon, sigmoid colon, rectum, and anus. To identify potentially malignant cells in the colon, scientists developed a CNN-classified CAD system. Multiple sets of colonoscopy images, up to and including 9995, were used to evaluate the capability of the CAD system to provide exhaustive findings for the entire colon. This study utilised real-time data from colonoscopies performed from January to October 2017. GoogleNet, CNN’s primary model, has an aggregate accuracy of 91.7% across 507 images. There are additional CNN models, but they were not examined in this study. Moreover, there is a substantial underutilisation of the numerous datasets that are accessible for this type of research.

## 2. Materials and Methods

This study’s primary objective is to develop a trustworthy system for the automated and human-reviewed detection and classification of colorectal cancer. The research suggests developing a CADx (Computer-Aided Diagnosis) [24] system for this purpose, which is EnsemDeepCADx. The proposed method analyses and interprets colorectal medical images using cutting-edge technology to expedite the diagnostic process. Figure 1 is a visual representation of the block diagram of the proposed system, which details the diagnostic procedure’s different stages. The EnsemDeepCADx system seeks to provide accurate and rapid colorectal cancer detection by combining automated analysis with human knowledge, which may contribute to enhanced patient outcomes and disease management.

### 2.1. Colonoscopy Medical Motion Images

During a colonoscopy, medical images are captured as the colonoscope is passed through each section of the intestine. To begin the procedure, the colonoscope is inserted into the rectum and cecum on the right side of the lower abdomen. The colonoscope is inserted through the right side of the abdomen and travels through the ascending colon, transverse colon, and descending colon. The colonoscope is used to access the sigmoid colon, which connects the lower left abdomen to the upper right. The connected camera captures high-resolution images as the colonoscope passes through each segment, enabling clinicians to carefully examine the colon’s walls for irregularities or diseases, such as polyps or fibroid tumours. These images are essential for the precise diagnosis and staging of colorectal cancer because they enable clinicians to identify and monitor the progression of abnormal tissue in the colon. In conclusion, the colonoscopy procedure stores medical images of the entire length of the colon, allowing doctors to detect and diagnose problems in this vital digestive organ.

#### Datasets

The colonoscopy medical image collections are accessible to anyone with internet access. Various online resources offer the CVC Clinic DB [25], Kvasir2 [26,27], and Hyper Kvasir [28,29] datasets, among others, for free download. There may be anywhere from two to twenty-three subclasses labelled for the lower and upper gastrointestinal tracts, depending on the extent of the dataset. CVC Clinic DB contains a total of 1640 images, equitably divided between polyps and non-polyps. Eight distinct classes are represented in the 8000 images comprising the Kvasir2 dataset. The Hyper Kvasir collection contains 10,672 images divided into 23 subclasses that represent the lower and upper digestive systems.

Classes are further organised and labelled into 24 classifications, resulting in a new mixed dataset named CKHK-22 that combines the previously mentioned datasets. The diverse collection of data contains 19,621 images. However, fourteen of these classifications are extremely unbalanced, which may hinder the performance of image classification techniques. Using the 24-class mixed CKHK-22 datasets, they discovered that the large bias towards these 14 classes resulted in a decline in accuracy and severe misclassification. By disregarding the 14 problematic classes, the authors of this study were able to improve the accuracy and efficacy of the system by concentrating on the most stable classes and images.

To surmount the limitations of the mixed dataset and enhance the performance and accuracy of image classification, the authors conducted experiments with the 10 most consistent classes from the dataset. The image collection used in this classification experiment is composed of 14,287 images equitably distributed across 10 balanced classes based on the colon component type. The distribution of the heterogeneous CKHK datasets is shown in Table 1. This approach increased the system’s adaptability to user requirements.

The CKHK-22 mixed dataset consists of 14,287 images from 10 distinct classes (bbps-0-1, bbps-2-3, cecum, dyed-lifted-polyps, dyed-resection-margins, non-polyps, polyps, pylorus, retroflex-stomach, and z-line). In terms of total images, class sizes vary from 653 for bbps-0-1 to 2150 for pylorus. The collection contains images from a variety of endoscopic techniques, including magnified narrow-band imaging (NBI) endoscopy, white light endoscopy (WLE), and chromoendoscopy. The dataset was constructed to facilitate the development of an EnsemDeepCADx system for image classification of colorectal cancer. Figure 2 depicts sample images from the CKHK-22 mixed dataset, which consists of 14,287 images in 10 classes.

### 2.2. Image Pre-Processing

Image pre-processing is crucial for improving image quality and classification model efficacy in colonoscopy images [30]. To facilitate the processing of images by deep learning models, it is customary to resize them to a uniform resolution of 224 by 224 pixels as part of the initial processing phase. Noise reduction is performed to prevent models from misclassifying data or omitting features. This includes removing imperfections such as grain, specks, and distortions. By colour correction, images’ colour balance can be made more consistent, which aids in classification. Normalisation helps standardise the pixel intensity values across the entire dataset, which is crucial for the efficient operation of various algorithms. Together, these pre-processing techniques optimise the classification quality of colonoscopy images.

#### 2.2.1. Google Cloud

The procedures for uploading the CKHK-22 dataset of colonoscopy medical images to Google Cloud for use with Co-lab [31] are as follows:Sign up for Google Cloud and create your first project to start;To begin storing the data, create a new container using the Cloud Storage interface;Using the Cloud Storage user interface or the command line utility, the CKHK-22 dataset may be added to the container;Ensure that the bucket is accessible to the public before uploading the data;Launch Google Authenticator and log in to your Co-lab account;Using gcsfuse, it can mount the container to Co-lab and access its contents as you would any other local disc;After mounting the container, the CKHK-22 dataset in Co-lab becomes as accessible as if it had been saved locally.

This article describes how to upload the CKHK-22 dataset of colonoscopy medical motion images for analysis and processing using Google Cloud and Co-lab.

#### 2.2.2. Image Augmentation

Image augmentation is a method for artificially increasing the size of a dataset by synthesising new images from the originals using a variety of techniques [32]. In the context of the colonoscopy dataset, image enhancement involves altering the relative proportions of:Resize: creates images that are 224 by 224 pixels in dimension;Noise reduction enhances image clarity by removing distracting ambient noise;Image colour correction is the process of standardising the image’s colour distribution to reduce variations;Using the zoom function, it can double or halve the image size;Images can be rotated by a maximum of 15 degrees;The horizontal flip function mirrors the image by horizontally rotating it.

These modifications enhance the aesthetic allure of the images and reduce the likelihood of over fitting during training. The augmented images are then added to the original dataset, expanding and diversifying it, which may improve the model’s precision and resilience. Using the enhanced dataset, the deep learning model is then trained on a platform such as Google Co-lab.

### 2.3. Train Test Split

The CKHK-22 dataset contains 14,287 images and 10 classes, as previously described. To facilitate model training and evaluation, the dataset is divided into a training set and a test set with a 70:30 division [33]. This indicates that 10,001 images (or 70% of the data) are used to train the model, whereas 4286 images (or 30% of the data) are reserved for evaluating the model on new data. The training set is used to teach the convolutional neural network to recognise patterns and make accurate predictions. To evaluate the generalisability of a model, its efficacy on new data (the test set) is evaluated. This technique prevents the model from being over fit to its training data, which could lead to substandard results when applied to novel data. By separating the dataset into training and test sets, the efficacy of convolutional neural networks for image classification can be evaluated more precisely on new data, allowing for a more accurate evaluation of the model’s performance.

#### 2.3.1. Extracting Grey Scale Features from Original Image Dataset

As a typical method in image processing, converting colour images to greyscale removes colour differences that may not be pertinent to the work at hand and compresses the data [34]. Greyscale conversion of the colour images in the CKHK-22 dataset may facilitate the classification process by eliminating colour-based information that is superfluous to the objective at concern. Greyscale images are produced by linearly combining the red, green, and blue (RGB) channels of the original colour image. In a greyscale image, the intensity values of individual pixels are calculated as follows:I = 0.2989 × R + 0.5870 × G + 0.1140 × B(1)
where, here, we symbolise the greyscale intensity and R, G, and B represent the original colour image’s red, green, and blue values. The brightness levels of the three colours are used to calculate these coefficients, with the red channel carrying the least weight and the green channel carrying the greatest.

The CKHK-22 colour images may be converted to greyscale by following these steps:Use a library for processing images, such as OpenCV, to read the source colour images;Apply the following formula to transform every image from the RGB colour space into the greyscale colour space;Create a new folder to store the greyscale images in so they may be used in the image classification model’s testing and training stages.

Since greyscale images only have one channel as opposed to colour images’ three, this transformation simplifies the dataset and speeds up the training process. This may also lessen the likelihood of the model being overfit to colour-specific traits that might not transfer well to other images. The pattern recognition and classification accuracy of convolutional neural networks (CNNs) can be enhanced by extracting greyscale image features from the CKHK-22 mixed dataset. By applying convolutional filters to the greyscale images, CNN can recognise edges, lines, and forms. The image is classified into one of ten classes after combining and supplying these features to entirely linked layers. By extracting greyscale image features, the model’s overall efficacy and performance on image classification tasks may be improved. The greyscale images are depicted in Figure 3. The conversion of the original colour images was used to generate the other type of CKHK-22 datasets.

#### 2.3.2. Extracting Local Binary Pattern Features from Greyscale Dataset

Local binary pattern (LBP) features are extracted from greyscale images to generate the LBP CKHK-22 dataset. The texture descriptor is then applied to the greyscale image, and a binary pattern is computed for each pixel based on the values of the adjacent pixels. The LBP operator establishes the pattern by comparing the central pixel’s intensity to that of its immediate companions. The value 1 is allocated to neighbouring pixels whose intensities are greater than or equal to the central pixel, while the value 0 is assigned otherwise. A pixel’s texture pattern is depicted by the decimal value obtained by translating its binary pattern [35]. The LBP features are extracted by sliding a 3 × 3 window across the greyscale CKHK-22 images and computing the LBP pattern for each pixel in the window. This procedure is repeated for each pixel, resulting in a new image in which each pixel has an LBP value. A histogram is then generated for each of the 16 × 16 non-overlapping segments that comprise the LBP image. The histograms constitute the LBP feature vector of the image.

The procedures for converting greyscale Images into LBP features are as follows:Create greyscale images that are typically 3 × 3 or 5 × 5 pixels in size;Determine the LBP of each pixel in the region by comparing its intensity to that of its neighbouring pixels;Replace the pixel’s original intensity value with a binary code representing the pattern of intensity differences between the central pixel and its companions;The LBP values of each pixel in the sub-region are added together to generate a singular LBP code;To generate a full set of LBP codes, the procedure must be repeated for each image’s subregion;To generate a feature vector that adequately characterises the image, the LBP codes are aggregated across the entire image using a histogram-based technique.

The resulting LBP CKHK-22 dataset contains ten classes and 14,287 greyscale images annotated with LBP features, 70:30 divided between a training set and a test set. These LBP features capture the texture information present in colonoscopy images, which is necessary for accurate image classification. The greyscale images are depicted in Figure 4. The conversion of the greyscale images was used to generate the other type of CKHK-22 datasets.

#### 2.3.3. Feature Fusion as New Dataset

The EnsemDeepCADx system for detecting colorectal cancer employs a multi-stage strategy that incorporates various features to effectively detect and identify polyps, with feature integration constituting an integral part of the system [36]. To extract features from the CKHK-22 dataset, they employ local binary patterns (LBPs), greyscale images, and raw RGB images. The result of combining these three datasets is the feature fusion dataset. Combining the full-size dataset with the greyscale and LBP datasets along the feature dimension yields a new dataset with three times as many images as the full-size dataset. The combined feature fusion dataset contains 42,861 images from the same 10 classes, with each of the three datasets contributing 14,287 images.

The derived features from the three datasets are combined into a singular feature vector during the fusion procedure. Combining several feature vectors by concatenating them along the feature dimension is one technique for fusing features [37]. Considering an input image size of 224 × 224, the original RGB dataset would consist of 3 × 224 × 224 = 150,528 features per image. Applying the same formula to the greyscale and LBP datasets would result in 50,176 and 2,650,816 features per image, respectively. Therefore, the final feature vector for each image in the feature fusion dataset would contain a total of 3 × 224 × 224 + 224 × 224 × 59 = 2,827,008 features.

The EnsemDeepCADx system relies significantly on the process of feature fusion, which unifies disparate data sources into a single representation. After feature fusion, the final dataset is typically divided 70:30 between training and testing datasets. The mere size of datasets such as ImageNet necessitates transfer learning at this time so that STM and SVM classifiers are then employed to classify the CNN-obtained features. CNNs can utilise the features learned by ensemble fusion CNN models. BiLSTM and SVM classifiers are then employed to classify the CNN-obtained features. Ensemble fusion CNNs trained with BiLSTM and SVM classifiers via transfer learning may enhance disease detection and diagnosis. All features extracted from the original CKHK-22 colour images, including grey and LBP images, are merged to create a new dataset termed the feature fusion dataset, as depicted in Figure 5.

### 2.4. Image Classification Using Ensemble Fusion CNNs

When it comes to image classification in EnsemDeepCADx, ensemble fusion CNNs are an excellent method for increasing the accuracy of predictions. This method produces an enhanced model by integrating multiple convolutional neural networks (CNNs) [38]. The aDaDR-22, aDaR-22, and DaRD-22 models represent a handful of instances of ensemble fusion CNNs applied to the CKHK-22 dataset.

The ADaDR-22 model consists of four previously trained networks: AlexNet, DarkNet-19, DenseNet-201, and ResNet-50. AlexNet is renowned for its accuracy and speed when processing complex data, while DarkNet-19 is admired for its speed and performance. DenseNet-201 is optimised for processing intricate features and correlations, whereas ResNet-50 is suitable for deep learning due to its skip connections. The ADaDR-22 model combines the most beneficial features of both varieties to achieve superior performance. aDaR-22 incorporates AlexNet, DarkNet-19, and ResNet-50, three models that were previously trained. This model resembles aDaDR-22, with the exclusion of DenseNet-201. This model’s objective is to simplify it while maintaining a high level of precision. DarkNet-19, ResNet-50, and DenseNet-201 are the DaRD-22 model’s construction elements. This model performs extremely well for complex data with associated properties. The DaRD-22 model integrates these three models to perform a wide range of image classification tasks efficiently and accurately.

During ensemble fusion, multiple CNN models are combined into a single, larger model. In order to diagnose colorectal cancer, the CADx system analyses and combines the most effective features from four pre-trained models into a single robust model. This method generates more precise and reliable forecasts than a single model could. Figure 6, Figure 7, Figure 8 and Figure 9 show the four pre-trained CNN models. AlexNet is composed of three fully connected layers and five convolutional layers. Rectified Linear Units (ReLU), a nonlinear activation function that accelerates training, are utilised. With the aid of local response normalisation and maximum aggregation, overfitting is minimised. AlexNet [39] was trained with 1.2 million images from 1000 classes, and it has a total of 60 million parameters. Multiple computer vision initiatives continue to employ this architecture. DarkNet-19 is a neural network architecture specifically designed for object recognition. It is a condensed version of the DarkNet-53 architecture found in the popular YOLOv3 object detector. DarkNet-19 [40] is composed of 19 convolutional layers based on the YOLOv2 architecture. To accomplish this, hybrid architecture composed of convolutional and max pooling layers is utilised. To facilitate gradient propagation and prevent gradients from dissipating, the network incorporates shortcut connections. DarkNet-19 employs batch normalisation and leaky ReLU activation functions to improve the training process and reduce overfitting. Compared to more complex models, DarkNet-19 uses fewer computational resources while maintaining high object identification accuracy. DenseNet-201 [41,42], upon which it is constructed, is praised for the dense interconnections that define its architecture. DenseNet-201 transmits feature maps downward through the layers, with each successively higher layer receiving the feature maps from all lower layers. By increasing gradient flow and utilising features, dense connections improve accuracy with fewer parameters. The dense structural elements consist of interconnected levels and systems. To reduce the spatial dimensions of the feature maps, they are divided into dense units and connected by transition layers such as a convolutional layer, a pooling layer, and a batch normalisation layer. After a global average pooling layer and a fully connected layer with a SoftMax activation function, the final layer of the network outputs the predicted class probabilities. ResNet50 [43,44] is a neural network with 50 convolutional, pooling, and fully connected layers. Residual connections enable the training of more complex networks and the learning of residual functions.

The purpose of this study is to integrate the beneficial features of multiple deep learning models. Specifically, the objective is to construct a model capable of extracting high-level features comparable to AlexNet, while employing the efficiency and lightweight design of DarkNet-19. In addition, the model intends to resolve the issue of DenseNet-201’s vanishing gradients and effectively manage the complexity of deep networks such as ResNet-50. The enhanced accuracy and performance of the resulting model are the result of its capacity to extract multiple features from complex datasets. There are numerous advantages to utilising multiple CNNs as opposed to one. Ensemble fusion CNNs uses the most accurate aspects of multiple models to enhance accuracy. Besides being more adaptable, these varieties are frequently more secure. Through ensemble fusion CNNs, overfitting, which occurs when a model becomes overly specific to the training data and underperforms on new data, can also be avoided. When applied to large, complex datasets such as CKHK-22, ensemble fusion CNNs such as aDaDR-22, aDaR-22, and DaRD-22 may significantly improve the accuracy of image classification. The detailed parameters of the ensemble fusion CNNs are described in Table 2.

#### 2.4.1. Bidirectional LSTM

Recurrent neural networks (RNNs) such as the Bidirectional Long Short-Term Memory (BILSTM) can process data sequences that contain both past and future information. BILSTMs are bidirectional RNNs, which means they can process information in both directions [45]. This makes the network a useful tool for predicting data sequences, as it can store both historical and future data. A BILSTM is constructed using two LSTM networks, one for processing the sequence forward and the other for processing the sequence reverse. The final result is the combined output of the two LSTMs [46]. This architecture is especially advantageous for applications requiring voice recognition, natural language processing, and image captioning, as it permits the network to acquire both immediate and deferred dependencies in the sequence. BILSTMs are utilised in the creation of image captions. The network is trained on a vast corpus of image–caption pairings for it to comprehend the underlying patterns and correlations between the visual elements of the images and the text description. Once the network has been trained, it can respond to any given image with a caption. The BiLSTM architecture is shown in Figure 10.

In the EnsemDeepCADx system, BILSTM may be utilised as an additional model to the ensemble fusion CNNs in order to improve classification accuracy. This is known as transfer learning. The BILSTM is pre-trained on a large dataset, such as ImageNet, to acquire generic image features. To further aid in the classification of colonoscopy images, these learned features may be transferred to the CADx system. Using the output features from the ensemble fusion CNNs, the BILSTM can discover temporal connections between image features. Due to the sequential nature of colonoscopy recordings, temporal data may be beneficial for identifying the images. The BILSTM may be beneficial for detecting early warning signs of cancer because it can record changes in image features over multiple time periods.

#### 2.4.2. Support Vector Machines

Support vector machines (SVMs), a type of supervised machine learning technique, can be utilised to perform classification and regression analysis [47]. SVMs can be used for classification and regression, which can both be linear or non-linear. SVMs aim to identify the optimal hyperplane for classifying a dataset into its constituent classes. SVMs employ n-dimensional (as many as the number of features in the dataset) plots of data points. The method then identifies the hyperplane that partitions the data into classes. Support vectors are the closest data points to the hyperplane, hence the name support vector machines.

Transfer learning can be used in tandem with multi-class SVMs by employing previously trained models as feature extractors [48]. The SVM algorithm may use as an input the features derived from the pre-trained models to classify data. Using pre-trained models to extract useful features can be advantageous when working with small datasets, as it eliminates the need for extensive training data. Transfer learning is utilised in conjunction with ensemble fusion CNNs and BILSTM in the EnsemDeepCADx system’s training on the CKHK-22 mixed dataset to extract features from the dataset. Finally, the generated feature vectors are fed to a multi-class SVM classifier for classification. This method permits the incorporation of temporal information into the feature vectors using BILSTM and the use of multiple pre-trained models to extract supplementary features. This method may improve the overall accuracy and efficacy of image classification for colorectal cancer.

### 2.5. The Classification Step

The procedure of the entire EnsemDeepCADx system classification process is described as follows:The input CKHK-22 mixed dataset contains 10 classes and 14,287 images;Before extracting local binary pattern (LBP) features, images from the CKHK-22 mixed dataset are converted to greyscale as part of the pre-processing step. The feature fusion dataset is created by combining these features with the RGB features from the original dataset;The CADx system’s ensemble fusion CNNs consist of four pre-trained CNN models: AlexNet, DarkNet-19, DenseNet-201, and ResNet-50. aDaDR-22, aDaR-22, and DaRD-22 are combined in three ways to create a more robust and accurate CNN model. The original RGB images and LBP features extracted from greyscale images are used to train ensemble CNNs on the CKHK-22 mixed dataset;The BILSTM recurrent neural network is capable of processing data sequences in both forward and reverse directions. In the EnsemDeepCADx system, the BILSTM is used as a transfer learning technique to enhance the efficacy of the ensemble CNNs. The resultant sequence of image features is then processed by the BILSTM layer, which receives input from the CNN ensemble. This method may improve the accuracy of a classification model by capturing temporal dependencies and correlations between visual features;SVMs are a family of machine learning algorithms used for classification and regression testing. SVMs are utilised as a post-processing stage in the EnsemDeepCADx system following the acquisition of classification results from an ensemble of CNNs and a BILSTM. An SVM classifier receives the results from the BILSTM layer and transforms the features into a higher dimension using a kernel function. The SVM classifier searches for the hyperplane that divides the input features into distinct categories in order to classify them. This technique helps improve the classification model’s accuracy by reducing false positives and enhancing class separation. This discovers how to partition data into the ten classes provided as input;Before performing transfer learning using ensemble CNNs, the final fully connected layer and SoftMax activation layer were eliminated from each CNN model. The feature maps produced by the final convolutional layer of each CNN were then provided to the BiLSTM layer. A total of 64 hidden units within the BiLSTM layer employed the tanh activation function. The output of the BiLSTM layer was input into a fully connected layer consisting of 32 hidden units and the ReLU activation function after final classification using a multi-Class SVM;Training the models: in this EnsemDeepCADx system, it can train the models using the pre-processed datasets and the hyperparameters specified for each model. The EnsemDeepCADx system can use techniques such as early stopping and learning rate scheduling to optimise the training process;Evaluating the models:Ie EnsemDeepCADx system can evaluate the models on the test set using metrics such as accuracy, precision, recall, and F1 score. In this system, the efficacy of a model is evaluated by producing ROC curves and calculating the area under the curve (AUC). To evaluate the efficacy of the trained models in real-world scenarios, an independent set of images from the CKHK-22 mixed dataset is used.Future images of colorectal cancer can be identified accurately using the completed EnsemDeepCADx system.

Figure 11 depicts the classification architecture, which integrates ensemble fusion CNNs with BiLSTM and multi-class SVM.

The classification procedure within this EnsemDeepCADx system consists of four experimental stages.

Stage 1:The CKHK-22 mixed image dataset is pre-processed to extract features from the original RGB images, as well as grey and LBP images, resulting in the creation of three new feature datasets. These three datasets are then merged to form a new feature fusion dataset.Stage 2:Three ensemble fusion CNN models—ADaDR-22, ADaR-22, and DaRD-22—are trained and tested with each of the four feature datasets (original, grey, LBP, and feature fusion).Stage 3:The three trained ensemble fusion CNN models are combined with BiLSTM models through transfer learning. The resulting models are then trained and tested with each of the four feature datasets.Stage 4:The three trained ensemble fusion CNN models are combined with BiLSTM and multi-class SVM models through transfer learning. The resulting models are then trained and tested with each of the four feature datasets.

The performance metrics obtained at each stage are then compared to determine which ensemble fusion CNN with BiLSTM and multi-class SVM provides the best recognition of colorectal cancer. The flow of the four stages is explained in detail in Figure 12.

The EnsemDeepCADx system developed in this study excels due to its novel combination of DaRD-22 ensemble fusion CNNs, bidirectional long short-term memory (BiLSTM), and support vector machines (SVM). This multimodal approach enables comprehensive analysis and diagnosis of colorectal cancer. The CKHK mixed dataset contains 14,287 images from nine distinct classes. We recognise the value of benchmarking against previous AI imaging methods, but it is essential to note that the primary objective of our research was to propose a novel method that combines ensemble learning and multimodal methodologies. Our research seeks to ascertain whether or not this combination improves the accuracy of colorectal cancer detection.

## 3. Experimental Setup

This article contains the experimental outcomes of the EnsemDeepCADx system. All testing was conducted on computers equipped with all the required hardware and software. This project’s software was chosen based on its compatibility with the system’s hardware and the task at hand. The studies utilised these software and hardware configurations because they were determined to be optimal for attaining the desired outcomes. The CADx system was deployed on a Dell Precision Tower T5810 machine outfitted with a 2.20 GHz Intel^®^ Xeon^®^ CPU core i7 E5-2630 processor and 32 GB of RAM. The NVIDIA Xp GPU accelerated the system’s processing power, making the deep learning models more accessible. The software stack included Keras and TensorFlow 2.7.0 as deep learning libraries, while the underlying operating system was Google Co-lab Pro+ running on Python 3.7.12. The ensemble fusion CNN models, BiLSTM models, and SVM models all used these libraries for training and testing. The EnsemDeepCADx system system’s hardware and software components were hand-picked to meet the demanding processing needs of the deep learning models and provide rapid, accurate classification.

This EnsemDeepCADx system relies heavily on its datasets. There are 10 classes for CKHK-22 in a mixed-dataset representation. Medical colonoscopy motion images are archived and made available for CNN training in all the various data classes. Each of the four feature types—original, grey level, LBP, and the merged dataset of original, grey level, and LBP—had their own training and testing sets inside the CKHK-22 dataset. Each feature type had a training set of 10,000 images and a testing set of 4287. There was a total of 42,861 images in the feature fusion dataset, including 30,000 training images and 12,861 testing images.

All datasets were subjected to transfer learning using BiLSTM and multi-class SVM, and ensemble fusion CNNs were used for experimental research. The hyperparameters were considered while planning the experiments. The batch size was 32, the learning rate was 0.0001, momentum was 0.9, and the number of epochs was 30. The optimiser was Adam. With a dropout rate of 0.5 and a batch size of 32, 30 epochs were used to train the BILSTM model. The multi-class SVM model utilised a linear kernel with a C value of 1.0. However, the batch size was increased to 128 to assure the seamless operation of the system on the 42,861-image feature fusion dataset. These hyperparameters were determined through a combination of empirical testing and comparative project data analysis.

Integrating multiple CNNs may result in the model becoming excessively complex, overlearning, and overfitting. Our research has taken several measures to resolve this issue. We have used regularisation techniques, including the dropout and weight decay, to reduce model complexity and prevent overfitting. These methods have the potential to reduce the danger of overlearning by employing regularisation constraints during training. Early stopping is a technique for terminating training early based on a model’s performance on a validation set. In order to prevent overlearning, training is terminated when no additional progress can be made or no negative effects can be observed. The hyperparameters of our model are optimal, establishing a balance between overfitting and underfitting. Changing the learning rate, sample size, and model architecture are a few of the numerous ways to avoid overlearning.

## 4. Results

The EnsemDeepCADx system was created to detect colorectal cancer by fusing together several variables from the CKHK-22 mixed dataset. Using the original, greyscale, LBP, and feature fusion datasets, three ensemble CNNs were merged with BiLSTM and multi-class SVM to boost the system’s accuracy.

Accuracy, precision, the F1 score, and recall were utilised as performance indicators to evaluate the effectiveness of the system. For each model set, the ROC and confusion curves were also mapped out. These measurements are the result of the following equations:Accuracy = (True Positives + True Negatives)/(True Positives + False Positives + True Negatives + False Negatives)(2)
Precision = True Positives/(True Positives + False Positives)(3)
Recall = True Positives/(True Positives + False Negatives) (4)
F1 Score = 2 × Precision × Recall/(Precision + Recall)(5)

The main resulting experiments were divided into the last three stages of the CADx system, with the findings from each phase compared to find the optimal ensemble for CADx-based colorectal cancer recognition.

### 4.1. Stage 1 Experimentation

After converting the original colour images to greyscale, the LBP transformation was applied to the CKHK-22 mixed dataset to generate LBP-featured images. Next, we fused the original, greyscale, and LBP featured images to create a new feature fusion dataset. All final three stages of the experiments employed these four datasets as inputs: the original, greyscale, LBP, and feature fusion. This is a crucial juncture for the EnsemDeepCADx system system, enabling crucial experiments to be conducted in subsequent stages. The subsequent investigations of the CADx system would not have been possible without the initial construction of the feature fusion dataset, which is the most important dataset for this system.

### 4.2. Stage 2 Experimentation: Ensemble Fusion CNNs

In the second stage of the experiment, the EnsemDeepCADx system employs three ensemble CNN models: ADaDR-22, ADaR-22, and DaRD-22. Each of the four CKHK-22 datasets (original, greyscale, LBP, and feature fusion) is used to train and evaluate these models; they contain a total of 10,000 training and 4287 testing images in three datasets and 30,000 training and 12,861 testing images in the feature fusion dataset, with 10 classes in each dataset.

Each ensemble CNN collects and calculates a variety of performance metrics, including accuracy, precision, F1 score, and recall. The system also contrasts the efficacy of the models and graphically displays the results. This stage of the experiment is crucial because it contrasts the accuracy of each model with different input data by evaluating the performance of each CNN ensemble with a variety of datasets. The results can be used to determine which CNN ensemble performs the best on the CKHK-22 dataset for detecting colorectal cancer. Table 3 presents the results of the Stage 2 experimental investigations of EnsemDeepCADx for ensemble CNNs using four datasets.

With a testing accuracy of 89%, DaRD-22 was the ensemble fusion CNN model with the maximum accuracy. This model’s precision was 90.78%, its recall was 89%, and its F1 score was 88.52%. This model’s training accuracy was 96.2%. The model with the lowest testing accuracy was ADaR-22, which had an accuracy of 86.33%. This model’s precision was 89.34%, its recall was 86.33%; and its F1 score was 85.66%. This model’s training accuracy was 94.22%.

The ADaR-22 model attained the highest accuracy for the Greyscale CKHK-22 dataset using ensemble fusion CNNs, with a precision of 85.47%, recall of 81.31%, F1 score of 82.07%, training accuracy of 91.33%, and testing accuracy of 82.07%, as shown in the table above. The DaRD-22 model obtained the lowest accuracy, with a precision of 81.95%, a recall of 80.66%, an F1 score of 79.57%, a training accuracy of 89.66%, and a testing accuracy of 81.60%.

Looking at the table, it appears that the DaRD-22 model in the LBP CKHK-22 dataset obtained the highest performance metrics. This model’s precision was 69.92%, its recall was 68.96%, and its F1 score was 67.56%. Training accuracy was 71.83%, while assessing accuracy was 68.96%. This indicates that the DaRD-22 model was able to effectively learn the LBP CKHK-22 dataset’s features and perform well when classifying the various classes. In contrast, the precision, recall, and F1 scores for the ADaDR-22 and ADaR-22 models were 64.4–68.74%, 65.31–66.03%, and 64.98–67.56%, respectively. The accuracy of training and testing was also inferior to the DaRD-22 model.

The ensemble fusion CNN model with the DaRD-22 architecture performed the best on the Feature Fusion CKHK-22 dataset, obtaining the highest precision (93.87%), recall (92.33%), and F1 score (91.3%). In addition, it had the maximum accuracy in training (95.46%) and testing (92.31%) compared to the other models. The ADaDR-22 model’s precision, recall, and F1 scores were 92.62%, 90.2%, and 89.5%, respectively. It had a lower training accuracy (92.06%) than the DaRD-22 model but a testing accuracy (90%) that was comparable. The ADaR-22 model performed the worst of the three, with precision, recall, and F1 scores of 91.56%, 89.69%, and 88.43%, respectively. In addition, it had the lowest accuracy during training (91.88%) and testing (89.69%) compared to the other models. Figure 13, Figure 14, Figure 15 and Figure 16 depict the comprehensive graphical analysis of the results of the second stage. Using ensemble fusion CNNs, the analysis revealed that DaRD-22 provided the highest level of accuracy in the Stage 2 experiment.

### 4.3. Stage 3 Experimentation: Ensemble Fusion CNNs + Multi-Class SVM

In the third stage of the investigation, the EnsemDeepCADx system integrates the three ensemble CNN models (ADaDR-22, ADaR-22, and DaRD-22) with a multi-class SVM to improve the colorectal cancer detection accuracy of the CKHK-22 dataset. Input for this stage is the feature fusion dataset, which has demonstrated superior performance in Stage 2.

The SVM functions as the final classifier in the EnsemDeepCADx system, receiving as input the fused features from the ensemble CNN models and producing the output. In Stage 2, the accuracy, precision, F1 score, recall, are computed for the combined ensemble CNN-SVM model, and the results are compared with the individual ensemble CNN models. This crucial stage of the experiment seeks to improve the accuracy of the EnsemDeepCADx system by integrating the strengths of ensemble CNNs and SVM for detecting colorectal cancer in the CKHK-22 dataset. Table 4 presents the results of the Stage 3 experimental investigations of EnsemDeepCADx for ensemble CNNs with multi-class SVM using four datasets.

Employing the original CKHK-22 datasets, the Stage 2 experiment compared the performance of the ensemble fusion CNNs when combined with a multi-class SVM using the original CKHK-22 datasets. Each model’s precision, recall, and F1 scores are displayed in the table above. The analysis revealed that the ADaR-22 model had the greatest precision value of 86.62% whereas the DaRD-22 model had the highest recall and F1 scores of 85.55% and 84.5%, respectively. The DaRD-22 model had the greatest training accuracy, at 88.67%, while the ADaDR-22 model had the highest testing accuracy, at 86.44%.

The ensemble fusion CNNs were combined with multi-class SVM and deployed to the greyscale CKHK-22 datasets in the Stage 2 experiment. The analysis results are presented in the table. DaRD-22 obtained the highest precision score with a value of 82.32%, followed by ADaDR-22 with 81.32% and ADaR-22 with 80.29%. DaRD-22 obtained the maximum recall score with a value of 80.62%, followed by ADaDR-22 with 79.91% and ADaR-22 with 77.79%. DaRD-22 obtained the highest F1 score with a value of 79.57%, followed by ADaDR-22 with 78.83% and ADaR-22% with 76.10%. DaRD-22 achieved the maximum training accuracy with 89.69%, followed by ADaDR-22 with 87.84% and ADaR-22 with 82.30%. DaRD-22 achieved the maximum testing accuracy with a value of 80.62%, followed by ADaDR-22 with 79.91% and ADaR-22 with 77.1%. Overall, DaRD-22 achieved the highest scores for most metrics, indicating that it performed the best among the three models on the greyscale CKHK-22 datasets.

In the Stage 2 experiment with the LBP CKHK-22 dataset, DaRD-22 attained the highest Precision, Recall, F1 Score, Training Accuracy, and Testing Accuracy among the three models, as shown in the table. DaRD-22 achieved a Precision of 69.46%, a Recall of 68.10%, an F1 Score of 67.50%, a Training Accuracy of 70.11%, and a Testing Accuracy of 68.10%. ADaR-22 achieved the lowest Precision, Recall, and F1 Score values, whereas ADaDR-22 achieved the lowest Training and Testing Accuracy values.

During the Stage 2 section of the experiment, the feature fusion CKHK-22 dataset was used to evaluate and contrast the capabilities of ensemble fusion CNNs in conjunction with multi-class SVM. The percentage of 91.59% was reached by DaRD-22, making it the system with the greatest recall, while the value of 92.63% was attained by DaRD-22, making it the system with the best accuracy. Additionally, DaRD-22 earned the highest F1 score, which was 90.48%. DaRD-22 attained the greatest training accuracy with a value of 90.89%, and it gained the best testing accuracy with a value of 91.59%. Both results were accomplished by DaRD-22.

The results indicate that DaRD-22 ensemble fusion CNNs combined with a multi-class SVM can enhance the performance of the models on the CKHK-22 datasets. Depending on the performance metric of concern, the best-performing model differs. Figure 17, Figure 18, Figure 19 and Figure 20 depict the comprehensive graphical analysis of the results of the third stage. Using ensemble fusion CNNs with multi-class SVM, the analysis revealed that DaRD-22 provided the highest level of accuracy in the Stage 3 experiment.

### 4.4. Stage 4 Experimentation: Ensemble Fusion CNNs + BiLSTM + Multi-Class SVM

In the final stage of the experiment, BiLSTM and multi-class SVM were merged with all three ensemble CNNs (ADaDR-22, ADaR-22, and DaRD-22). All four of the CKHK-22 featured datasets (original, greyscale, LBP, and feature fusion) were used in the model’s training and testing, with metrics including accuracy, precision, F1 score, and recall serving as measures of performance. The success of the EnsemDeepCADx system in identifying colorectal cancer utilising ensemble CNNs, BiLSTM, and multi-class SVM was determined by the outcomes of this stage. The results of this stage provided insight into the optimal mix of these models for boosting the system’s precision. Table 5 presents the results of the Stage 4 experimental investigations of EnsemDeepCADx for ensemble CNNs with BiLSTM and multi-class SVM using four datasets.

In the final phase of the experiment with the original CKHK-22 image datasets, DaRD-22 ensemble fusion CNNs achieved the highest values for precision, recall, F1 score, training accuracy, and testing accuracy, with a precision of 95.31%, a recall of 94.9%, an F1 score of 93.4%, a training accuracy of 98.64%, and a testing accuracy of 95.96%. ADaDR-22 ensemble fusion CNNs attained the lowest values, with a precision of 89.92%, a recall of 93.47%, an F1 score of 86.76%, a training accuracy of 97.74%, and a testing accuracy of 93.47%. ADaR-22 ensemble fusion CNNs achieved 92.12% precision, 91.58% recall, an F1 score of 85.61%, a training accuracy of 96.95%, and a testing accuracy of 91.58%.

The DaRD-22 model achieved the highest precision, recall, and F1 score for the ensemble fusion CNNS + BLSTM + multi-class SVM applied to the greyscale CKHK-22 dataset, with values of 90.46, 88.79, and 87.62, respectively. On the other hand, the ADaR-22 model obtained the lowest values for these metrics, with precision, recall, and F1 score values of 84.92%, 83.09%, and 82.57%, respectively. The DaRD-22 model also obtained the highest training and testing accuracy with values of 95.56% and 88.79%, respectively.

DaRD-22 had the highest ensemble fusion CNNs + BLSTM + multi-class SVM values for the LBP CKHK-22 datasets, with a precision of 75.11%, recall of 73.5%, F1 score of 72.67%, training accuracy of 75.89%, and testing accuracy of 73.5%. ADaDR-22 had the lowest values, with a precision of 70.67%, recall of 69.92%, F1 score of 67.41%, training accuracy of 72.56%, and testing accuracy of 69.92%.

The following is an analysis of the highest and lowest ensemble fusion CNNs + BLSTM + multi-class SVM values for the feature fusion CKHK-22 datasets. DaRD-22 obtained the highest accuracy with a value of 96.98%. This indicates that 96.98% of all predicted positive cases were in fact positive. DaRD-22 obtained the highest recall with a score of 97.12%. This indicates that 97.12% of all actual positive cases were accurately identified as positive. DaRD-22 obtained the greatest F1 score, with a value of 95.98%. This is the harmonic mean of precision and recall, balancing the two metrics. DaRD-22 obtained the maximum training accuracy with a value of 98.72%. This indicates that the model correctly categorised 98.72% of the training dataset. DaRD-22 obtained the greatest testing accuracy with a value of 97.89%. This indicates that the model accurately classified 97.89% of the test dataset. On the other hand, ADaR-22 obtained the lowest values for all metrics. Nonetheless, even the lowest values are comparatively high, indicating that the model performs well in general.

Figure 21, Figure 22, Figure 23 and Figure 24 depict the comprehensive graphical analysis of the results of the third stage. Using ensemble fusion CNNs with BiLSTM and multi-class SVM, the analysis revealed that DaRD-22 provided the highest level of accuracy in the Stage 4 experiment.

The DaRD-22 classifier, an ensemble fusion CNN model with BiLSTM and multi-class SVM classifiers, scored the greatest accuracy across all metrics, with a testing accuracy of 97.89%, according to the findings of experimental investigations conducted on the feature fusion CKHK-22 dataset. The model also performed well in terms of accuracy in identifying positive instances and maintaining a low false positive rate (measured by precision, recall, and F1 score). The excellent 98.72% training accuracy also shows that the model is appropriate for the training data and can generalise to novel, unknown test data with ease. The DaRD-22 model outperformed BiLSTM and multi-class SVM classifiers on the feature fusion CKHK-22 dataset, making it the best model for reliably detecting colorectal cancer in this EnsemDeepCADx system.

Presented in Table 6 are the performance metrics for ensemble fusion CNN-DarD-22 with BiLSTM and multi-class SVM, utilising the feature fusion CKHK-22 mixed dataset. Each row in the table corresponds to a specific class of polyps or non-polyp regions in the colon, and each column provides a different performance metric for that class.

Precision is the proportion of true positive predictions for a given class out of all positive predictions for that class. In other words, precision measures the proportion of correctly identified instances of a given class out of all instances predicted as that class. The classes with precision above 0.9 are bbps-0-1, bbps-2-3, non-polyps, pylorus, retroflex-stomach, and z-line.

Recall assesses the percentage of true positive predictions for a given class out of all actual instances of that class in the test dataset. In other words, recall measures the proportion of correctly identified instances of a given class out of all actual instances of that class. The classes with recall above 0.9 are bbps-0-1, bbps-2-3, cecum, non-polyps, pylorus, retroflex-stomach, and z-line.

The F1 score is the harmonic mean of precision and recall. It is a single metric that combines precision and recall into one number. The classes with an F1 score above 0.9 are bbps-0-1, bbps-2-3, non-polyps, pylorus, retroflex-stomach, and z-line.

Support refers to the number of test images that belong to a particular class. The classes with the highest support in this dataset are polyps and dyed-lifted-polyps with 2604 and 1803 images, respectively. However, the highest-performing classes based on precision, recall, and F1 score are bbps-0-1, bbps-2-3, non-polyps, pylorus, retroflex-stomach, and z-line.

Overall, the results suggest that the ensemble fusion CNN-DarD-22 using the feature fusion CKHK-22 mixed dataset can accurately identify polyps and non-polyp regions in the colon, with several classes exhibiting high precision, recall, and F1 score. The analysed performance metrics are shown in Figure 25.

From the foregoing, we may deduce that this classifier’s confusion matrix will contain many true positives and true negatives for the classes with high recall values, and many false negatives for the classes with low recall values. Figure 25 illustrates the confusion matrix for the DaRD-22 with BiLSTM and multi-class SVM performance of the ensemble fusion CNN on the CKHK-22 dataset. More insight into the classifier’s performance might be gained with the use of a thorough confusion matrix, which would show the real number of true positives, false positives, true negatives, and false negatives for each class. The performance metric recall indicates how many true positive instances were accurately labelled as such by the classifier. The following may be inferred from the table of recall values: it turns out that “bbps-0-1” (0.98), “bbps-2-3” (0.99), “cecum” (0.99), “polyps” (0.82), “pylorus” (1.00), “retroflex-stomach” (0.99), and “z-line” (0.99) had the greatest recall values. These are the categories for which a large percentage of true positives were properly identified by the classifier. Recall values for “dyed-lifted-polyps” (92.2%) and “dyed-resection-margins” (43.3%) were the lowest. These are the categories where the classifier produced a larger number of false negatives because it incorrectly classified a large percentage of true positives.

The ROC curve is a graphical representation of a binary classifier system’s performance as its discrimination threshold is altered. Figure 26 depicts the confusion matrix and Figure 27 depicts the ROC curve for DaRD-22 using the CKHK-22 feature fusion dataset. The True Positive Rate (TPR) is plotted on the *y*-axis and the False Positive Rate (FPR) is plotted on the *x*-axis. AUC (Area Under the Curve) is a metric that assesses the classification system’s overall performance. In this instance, the AUC value is 0.9882, indicating that the efficacy of the classifier system is very high. It indicates that the system can effectively differentiate between positive and negative samples. Since the AUC value is close to 1, it can be inferred that there is no misclassification in any class and that the accuracy of all classes is greater than 0.96. The EnsemDeepCADx system is a powerful instrument for detecting and diagnosing colorectal cancer, as evidenced by its AUC of 0.9882 and outstanding accuracy values in all 10 classes.

## 5. Discussion

Several studies have proposed CADx systems for the diagnosis of colorectal cancer, each employing a unique set of methodologies and models.

On the CVC Clinic DB dataset, Liew et al. (2021) [11] used an ensemble classifier approach with ResNet50 + Adaboost, AlexNet, GoogLeNet, and VGG-19 models to achieve an accuracy of 97.91%. Their method’s execution time was 2.5 h;Omneya Attallah et al. (2021) [17] obtained an accuracy of 97.3% and 99.7% on the Kvasir2 and Hyper Kvasir datasets, respectively, using the GastroCADx method with AlexNet, DarkNet19, ResNet50, DenseNet-201, DWT, DCT, and SVM. The duration of execution for both datasets was three hours;Maryem Souaidi et al. (2022) [21] applied the MP-FSSD technique with VGG16 and feature Fusion Module to the CVC Clinic DB and WCE datasets. They obtained an accuracy of 91.56 percent in 2.5 h of execution;Pallabi Sharma et al. (2022) [49] utilised an ensemble classifier technique with ResNet101, GoogleNet, and XceptionNet models on the CVC Clinic DB and Aichi Medical Dataset. They obtained a 98.3% accuracy rate in 2.45 h of execution;Nisha J.S. et al. (2022) [50] applied the DP-CNN technique with the Dual Path CNN model to the CVC Clinic DB and ETIS-Larib datasets, achieving a 99.6% accuracy. Their method’s execution time was two hours;ColoRectalCADx was developed by Akella S. Narasimha Raju et al. [51] using ResNet-50V2, DenseNet-201, VGG16, LSTM, and SVM models on Hyper Kvasir Balanced and Mixed Dataset Balanced. They attained 98.91% and 96.13% accuracy with execution times of 2.15 and 2.10 h, respectively;EnsemDeepCADx, the proposed model (2023), employed Ensemble CNN DaRD-22, BLSTM, and SVM with feature fusion on the CKHK-22 mixed dataset. The accuracy was 97.89% and the execution time was 2 h.

In contrast, the EnsemDeepCADx system proposed in this study utilised the DaRD-22 ensemble fusion CNN along with BLSTM and SVM to achieve a 97.89% accuracy on the CKHK-22 mixed feature fusion dataset. This study demonstrates the potential for deep learning and transfer learning to improve the performance of CADx systems for the early detection of colorectal cancer. Table 7 compares the proposed EnsemDeepCADx system for 2023 to colorectal cancer procedures in 2021 and 2022, as well as their respective descriptions.

The comparison and discussion emphasise the diverse methodologies and approaches used by the various studies to develop CADx systems for the diagnosis of colorectal cancer. Even though a number of studies have produced exceptional precision, there is still ample opportunity for growth. This study demonstrates positive outcomes for the proposed CADx system, and future research could build on this foundation by investigating complementary models and techniques. The progress made in the area of colorectal cancer detection is graphically shown in Figure 28.

An “explainable AI” system is one that can explain its logic behind a prediction or action in a manner that a human can understand. To diagnose colorectal cancer, EnsemDeepCADx employs an ensemble of convolutional neural networks (CNNs; particularly, DaRD-22), bidirectional long short-term memory (BLSTM), and support vector machines (SVM) with feature fusion. One of the main advantages of the EnsemDeepCADx system is its interpretability. CNNs enable the system to learn and extract information relevant for diagnosing colorectal cancer from colonoscopy pictures. The ensemble approach boosts the system’s performance even more by combining many models and relying on their combined expertise. The BiLSTM component is incorporated to improve the system’s interpretability because of its ability to capture temporal linkages and sequential patterns within image data. This enables the computer to take into consideration the illness’s history and context, resulting in more exact estimates. Furthermore, the SVM algorithm is employed as a decisive element in defining the many types of colorectal cancer. This opens the path for more accurate cancer and other illness diagnosis. Feature fusion in EnsemDeepCADx combines data from many image representations, including colour, greyscale, and local binary pattern (LBP) pictures. We can better capture the intricacies of the underlying data and perform a more exact analysis by integrating these various attributes.

When these variables are integrated, EnsemDeepCADx’s prediction accuracy for colorectal cancer jumps to an astounding 97.89%. The system’s interpretability instils trust in the decision-making process among healthcare providers. Clinicians can better interact with patients and back up their own assessment if they understand what goes into the system’s predictions.

Finally, the EnsemDeepCADx system combines explainable AI ideas by using interpretable components such as CNNs, BLSTMs, and SVMs, as well as feature fusion approaches. These design changes were taken in order to increase the system’s utility to clinicians in the identification of colorectal cancer by making it more open and offering more relevant reasons for its forecasts.

## 6. Conclusions and Future Work

Employing a combination of the ADaDR-22, ADaR-22, and DaRD-22 ensemble fusion CNNs, the EnsemDeepCADx system has been shown to increase diagnostic accuracy in the identification of colorectal cancer. Information from colonoscopy pictures is extracted using deep learning architectures such as AlexNet, DarkNet-19, DenseNet-201, and ResNet-50. The EnsemDeepCADx system is evaluated using the CKHK-22 mixed dataset, which includes colour, greyscale, and LBP image datasets to showcase the system’s adaptability. By using a feature fusion method to combine the collected characteristics, the system performs a comprehensive analysis of the provided input. The EnsemDeepCADx system has a top testing accuracy of 97.12% thanks in large part to the DaRD-22 ensemble CNN fusion, BILSTM, and multi-class SVM. The system’s potential for early and accurate detection of colorectal cancer is shown by this degree of accuracy. One of the numerous advantages of the EnsemDeepCADx system is how quickly data can be processed. Colorectal cancer may now be identified in as little as 2 h thanks to this cutting-edge technology. This expedited process increases confidence that the system will provide useful results in a reasonable amount of time. Additional studies into various deep learning approaches and parameter optimisation are emphasised as a means to further enhance the system’s performance. The value of utilising ensemble fusion CNNs to boost colorectal cancer detection accuracy and throughput is also emphasised.

In conclusion, the EnsemDeepCADx system’s utilisation of ensemble fusion CNNs and integration of BILSTM and SVM shows potential in the domain of colorectal cancer detection. The system’s ability to include cutting-edge deep learning algorithms and feature fusion approaches while maintaining an efficient processing time of 2 h increases the likelihood that it will significantly contribute to early detection and improved patient outcomes.

Even though the current EnsemDeepCADx system has demonstrated some success, it could be improved further. Future research may focus on advanced deep learning techniques, such as generative adversarial networks (GANs) for data augmentation and attention mechanisms for enhanced feature extraction. The system may utilise data from other medical imaging modalities, such as magnetic resonance imaging (MRI) or computed tomography (CT) examinations, to enhance its diagnostic capabilities. Future research may also concentrate on augmenting the technology for use in real time during colonoscopies, thereby facilitating instantaneous feedback and, potentially, a more rapid diagnosis and treatment. Ongoing research and development of CADx systems for the early diagnosis of colorectal cancer will likely result in improved patient outcomes in the long term.

## Figures and Tables

**Figure 1 bioengineering-10-00738-f001:**
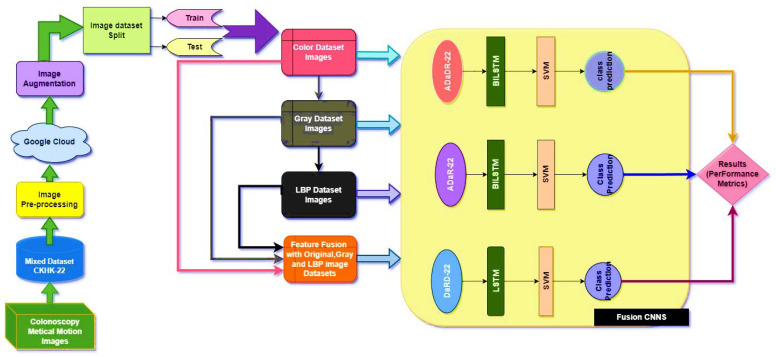
Block diagram of the CADx system.

**Figure 2 bioengineering-10-00738-f002:**
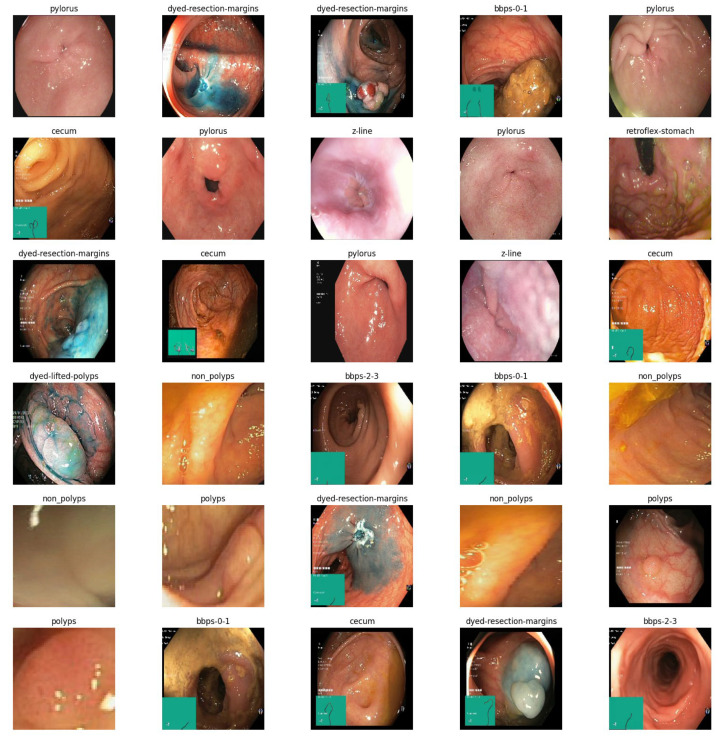
CKHK-22 mixed dataset original images.

**Figure 3 bioengineering-10-00738-f003:**
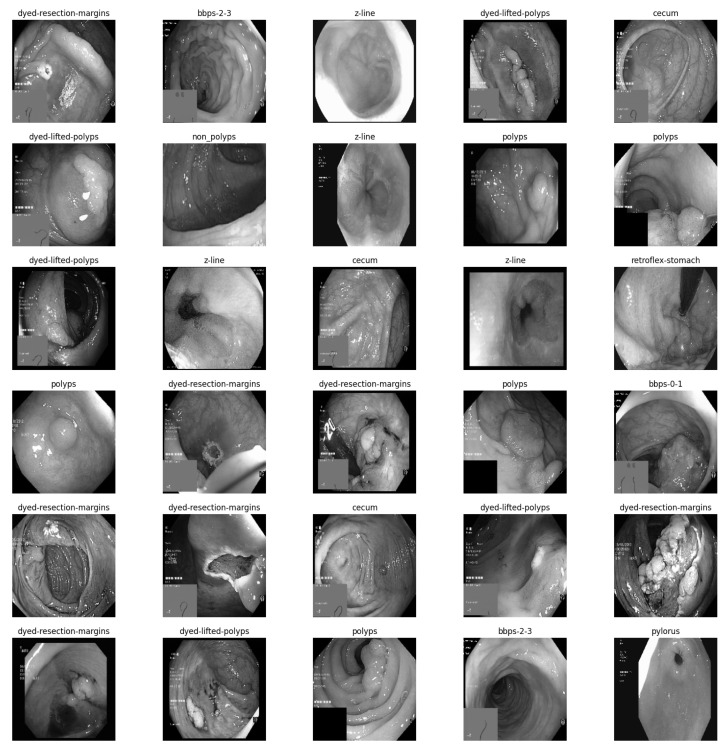
CKHK-22 mixed dataset greyscale images.

**Figure 4 bioengineering-10-00738-f004:**
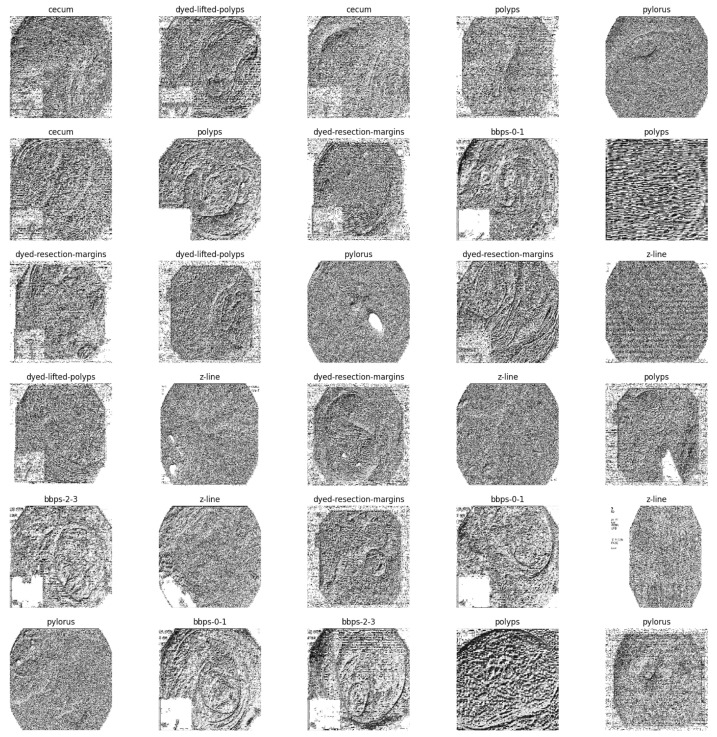
CKHK-22 mixed dataset Local Binary Pattern (LBP) images.

**Figure 5 bioengineering-10-00738-f005:**
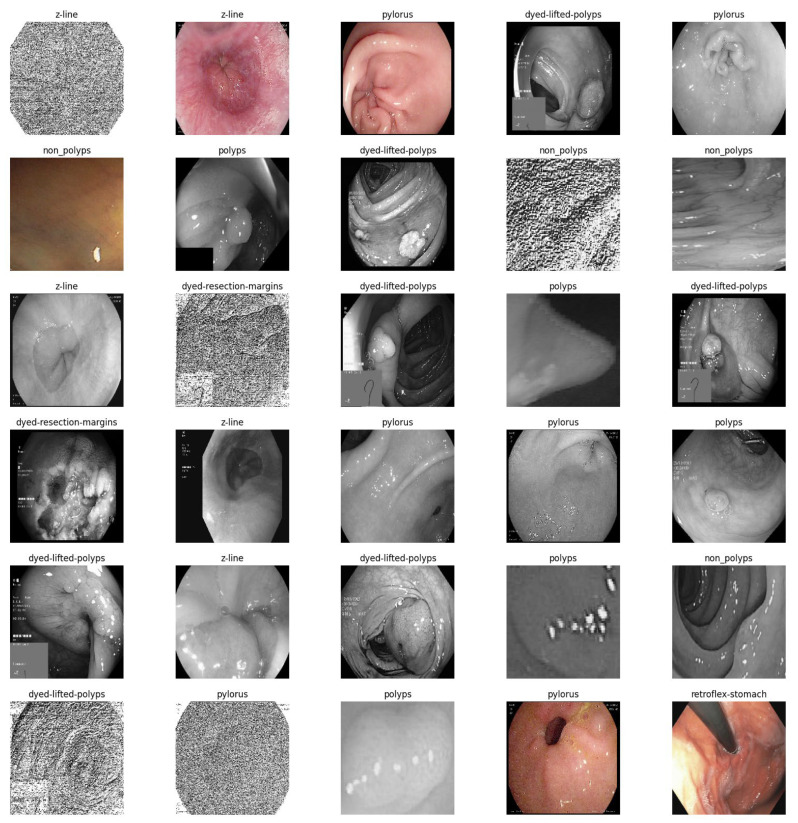
CKHK-22 mixed dataset feature fusion images.

**Figure 6 bioengineering-10-00738-f006:**
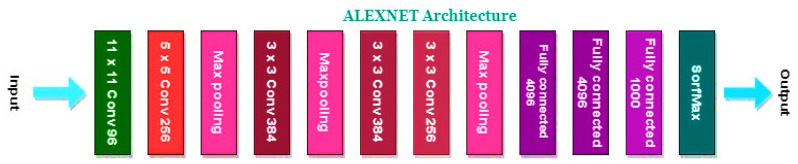
Architecture of the Convolutional Neural Network (CNN) AlexNet.

**Figure 7 bioengineering-10-00738-f007:**
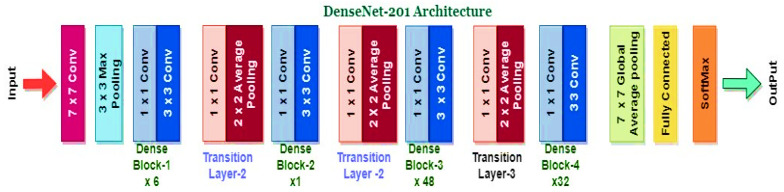
Architecture of the CNN DarkNet-19.

**Figure 8 bioengineering-10-00738-f008:**
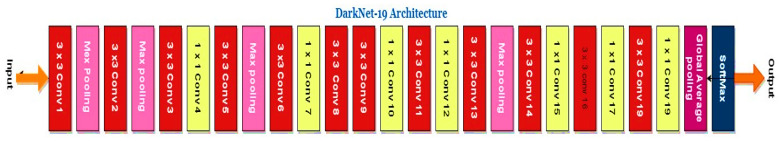
Architecture of the CNN DenseNet-201.

**Figure 9 bioengineering-10-00738-f009:**
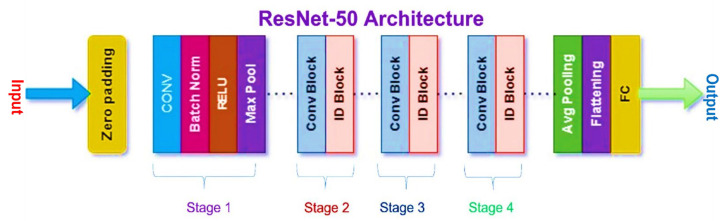
Architecture of the CNN ResNet-50.

**Figure 10 bioengineering-10-00738-f010:**
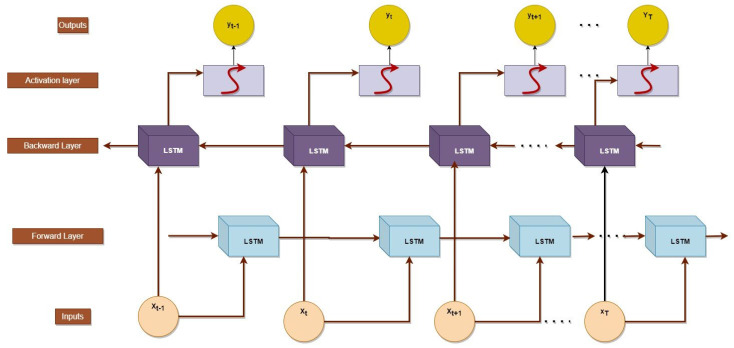
Architecture ofBidirectional Long Short-Term Memory (BiLSTM.)

**Figure 11 bioengineering-10-00738-f011:**
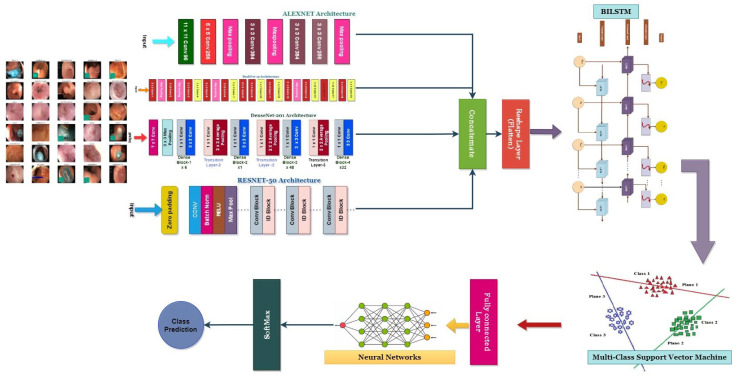
Classification architecture of the ensemble fusion CNNs combined with BiLSTM and multi-class Support Vector Machine (SVM).

**Figure 12 bioengineering-10-00738-f012:**
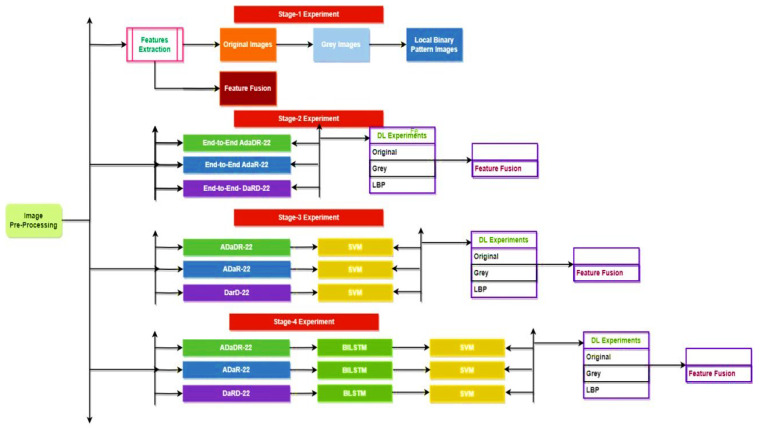
CADx system experimental setup flow in four stages.

**Figure 13 bioengineering-10-00738-f013:**
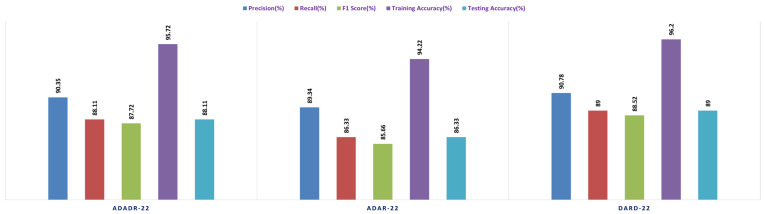
Stage 2 experiment analysis of ensemble CNNs with the original CKHK-22 dataset.

**Figure 14 bioengineering-10-00738-f014:**
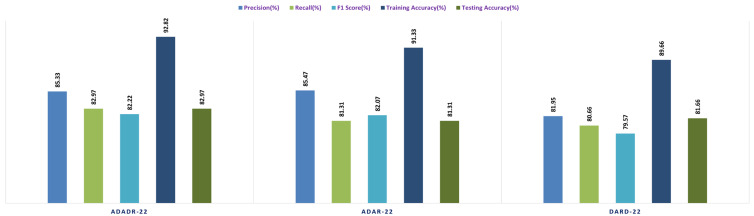
Stage 2 experiment analysis of ensemble CNNs with the greyscale CKHK-22 dataset.

**Figure 15 bioengineering-10-00738-f015:**
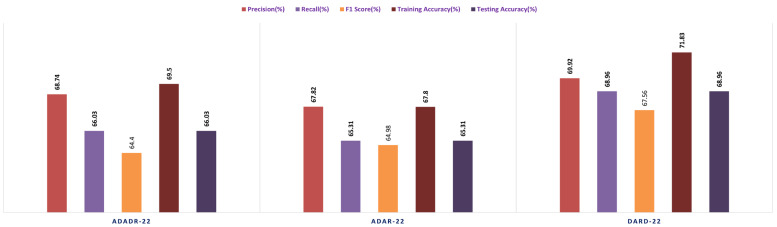
Stage 2 experiment analysis of ensemble CNNs with the LBP CKHK-22 dataset.

**Figure 16 bioengineering-10-00738-f016:**
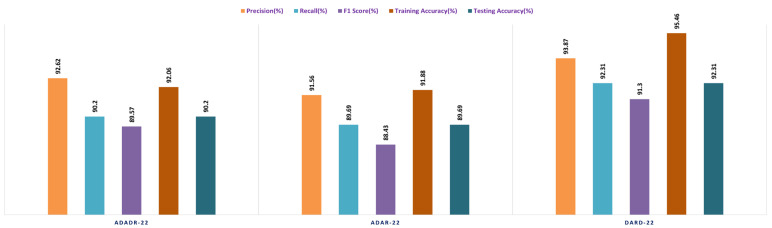
Stage 2 experiment analysis of ensemble CNNs with the feature fusion CKHK dataset.

**Figure 17 bioengineering-10-00738-f017:**
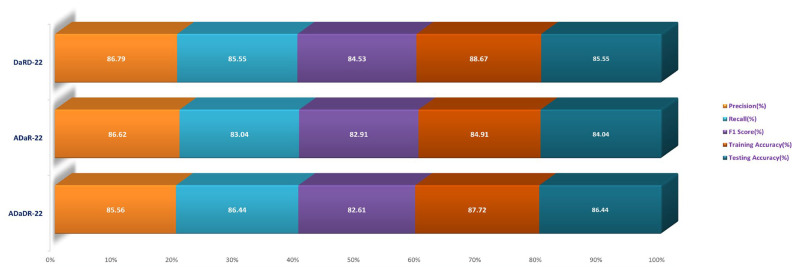
Stage 3 experiment analysis of ensemble CNN + multi-class SVM with the original CKHK- 22 dataset.

**Figure 18 bioengineering-10-00738-f018:**

Stage 3 experiment analysis of ensemble CNN + multi-class SVM with the greyscale CKHK-22 dataset.

**Figure 19 bioengineering-10-00738-f019:**
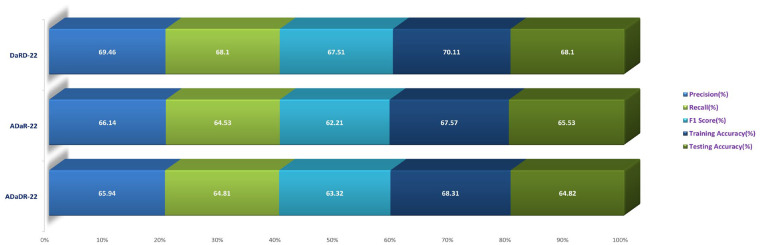
Stage 3 experiment analysis of ensemble CNN + multi-class SVM with the LBP CKHK-22 dataset.

**Figure 20 bioengineering-10-00738-f020:**
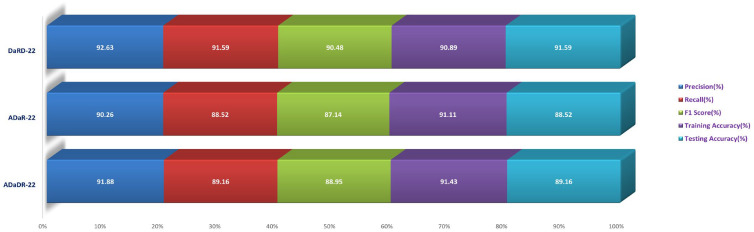
Stage 3 experiment analysis of ensemble CNN + multi-class SVM with the feature fusion CKHK dDataset.

**Figure 21 bioengineering-10-00738-f021:**
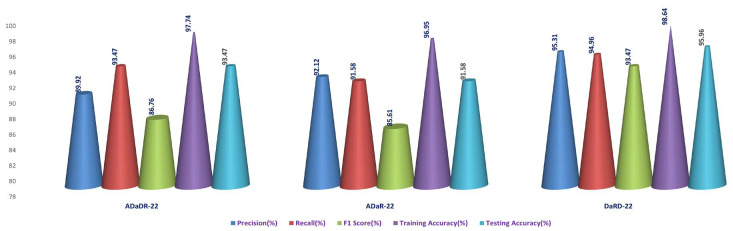
Stage 4 experiment analysis of ensemble CNN + BiLSTM + multi-class SVM with the original CKHK-22 dataset.

**Figure 22 bioengineering-10-00738-f022:**
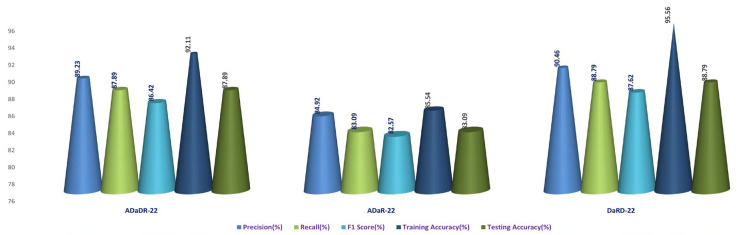
Stage 4 experiment analysis of ensemble CNN + BiLSTM + multi-class SVM with the greyscale CKHK-22 dataset.

**Figure 23 bioengineering-10-00738-f023:**
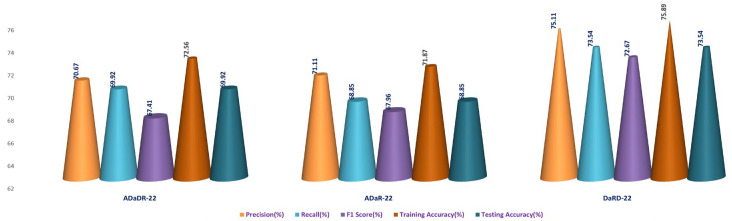
Stage 4 experiment analysis of ensemble CNN + BiLSTM + multi-class SVM with the LBP CKHK-22 dataset.

**Figure 24 bioengineering-10-00738-f024:**
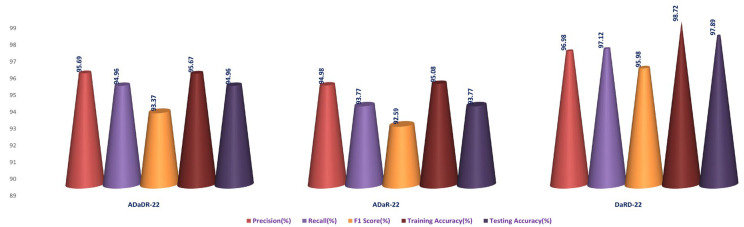
Stage 4 experiment analysis of ensemble CNN + BiLSTM + multi-class SVM with the feature fusion CKHK-22 dataset.

**Figure 25 bioengineering-10-00738-f025:**
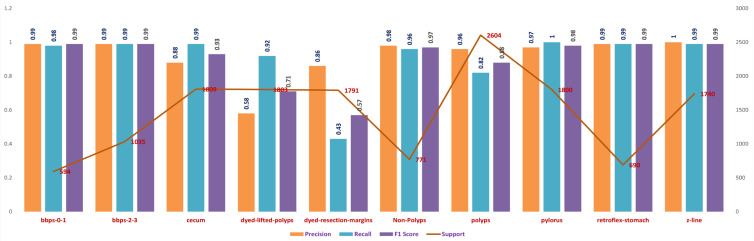
Performance metrics of individual classes of the feature fusion CKHK-22 dataset.

**Figure 26 bioengineering-10-00738-f026:**
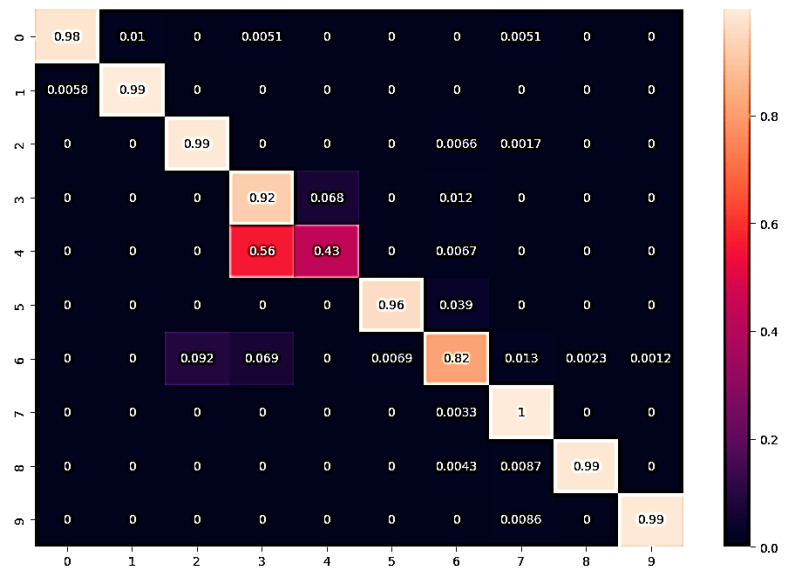
Confusion matrix of the best model in the EnsemDeepCADx system.

**Figure 27 bioengineering-10-00738-f027:**
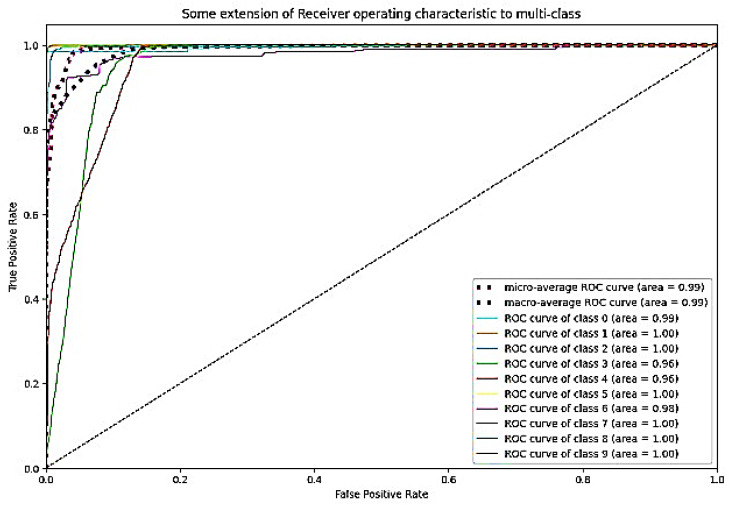
ROC curve of the best model in the EnsemDeepCADx system.

**Figure 28 bioengineering-10-00738-f028:**
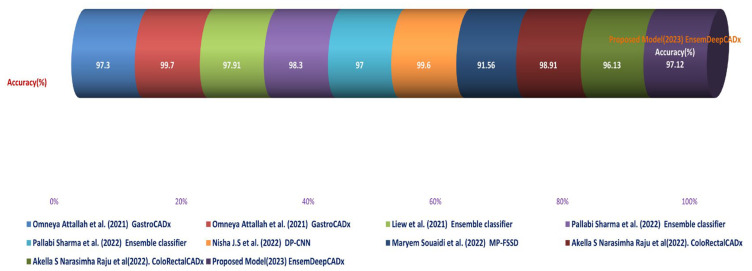
Comparison of the state-of-the-art methods for colorectal cancer detection.

**Table 1 bioengineering-10-00738-t001:** Description of dataset images.

Classes	CKHK-22 Mixed Dataset	Images
0	**bbps-0-1**	653
1	**bbps-2-3**	1148
2	**cecum**	2009
3	**dyed-lifted-polyps**	2003
4	**dyed-resection-margins**	1990
5	**non-polyps**	818
6	**polyps**	818
7	**pylorus**	2150
8	**retroflex-stomach**	765
9	**z-line**	1933
	**Total images in dataset**	14,287

**Table 2 bioengineering-10-00738-t002:** Number of parameters of CNNs.

CNN Architecture Models	Introduced Year	Total Params	Trainable Params	Non-Trainable Params	Layers
**ADaDR-22**	2022	89,491,098	47,570,314	41,920,784	293
**ADaR-22**	2022	70,062,380	46,463,580	23,598,800	92
**DaRD-22**	2022	61,401,236	19,501,588	41,899,648	270

**Table 3 bioengineering-10-00738-t003:** Performance metrics comparison of ensemble CNNs in the Stage 1 experiment.

Datasets	Ensemble Fusion CNNs	Precision (%)	Recall (%)	F1 Score (%)	Training Accuracy (%)	Testing Accuracy (%)
Original CKHK-22	**ADaDR-22**	90.35	88.11	87.72	95.72	88.11
Datasets	**ADaR-22**	89.34	86.33	85.66	94.22	86.33
	**DaRD-22**	90.78	89	88.52	96.2	89
**Datasets**	**Ensemble Fusion CNNs**	**Precision (%)**	**Recall (%)**	**F1 Score (%)**	**Training Accuracy (%)**	**Testing Accuracy (%)**
Greyscale	**ADaDR-22**	85.33	82.97	82.22	92.82	82.97
CKHK-22 Datasets	**ADaR-22**	85.47	81.31	82.07	91.33	81.31
	**DaRD-22**	81.95	80.66	79.57	89.66	81.66
**Datasets**	**Ensemble Fusion CNNs**	**Precision (%)**	**Recall (%)**	**F1 Score (%)**	**Training Accuracy (%)**	**Testing Accuracy (%)**
LBP CKHK-22	**ADaDR-22**	68.74	66.03	64.4	69.5	66.03
Datasets	**ADaR-22**	67.82	65.31	64.98	67.8	65.31
	**DaRD-22**	69.92	68.96	67.56	71.83	68.96
**Datasets**	**Ensemble Fusion CNNs**	**Precision (%)**	**Recall (%)**	**F1 Score (%)**	**Training Accuracy (%)**	**Testing Accuracy (%)**
Feature Fusion	**ADaDR-22**	92.62	90.2	89.57	92.06	90.2
CKHK-22 Datasets	**ADaR-22**	91.56	89.69	88.43	91.88	89.69
	**DaRD-22**	93.87	92.31	91.3	95.46	92.31

**Table 4 bioengineering-10-00738-t004:** Performance metrics comparison of ensemble CNNs + multi-class SVM in the Stage 3 experiment.

Datasets	Ensemble Fusion CNNs	Precision (%)	Recall (%)	F1 Score (%)	Training Accuracy (%)	Testing Accuracy (%)
**Original CKHK-22 Datasets**	**ADaDR-22**	85.56	86.44	82.61	87.72	86.44
	**ADaR-22**	86.62	83.04	82.91	84.91	84.04
	**DaRD-22**	86.79	85.55	84.53	88.67	85.55
**Datasets**	**Ensemble Fusion CNNs**	**Precision (%)**	**Recall (%)**	**F1 Score (%)**	**Training Accuracy (%)**	**Testing Accuracy (%)**
**Greyscale CKHK-22 Datasets**	**ADaDR-22**	81.32	79.91	78.83	87.84	79.91
	**ADaR-22**	80.29	77.79	76.1	82.32	77.19
	**DaRD-22**	82.32	80.62	79.57	89.69	80.62
**Datasets**	**Ensemble Fusion CNNs**	**Precision (%)**	**Recall (%)**	**F1 Score (%)**	**Training Accuracy (%)**	**Testing Accuracy (%)**
**LBP CKHK-22 Datasets**	**ADaDR-22**	65.94	64.81	63.32	68.31	64.82
	**ADaR-22**	66.14	64.53	62.21	67.57	65.53
	**DaRD-22**	69.46	68.1	67.51	70.11	68.1
**Datasets**	**Ensemble Fusion CNNs**	**Precision (%)**	**Recall (%)**	**F1 Score (%)**	**Training Accuracy (%)**	**Testing Accuracy (%)**
**Feature Fusion CKHK-22 Datasets**	**ADaDR-22**	91.88	89.16	88.95	91.43	89.16
	**ADaR-22**	90.26	88.52	87.14	91.11	88.52
	**DaRD-22**	92.63	91.59	90.48	90.89	91.59

**Table 5 bioengineering-10-00738-t005:** Performance metrics comparison of ensemble CNNs +BiLSTM + multi-class SVM in the Stage 4 experiment.

Datasets	Ensemble Fusion CNNs	Precision (%)	Recall (%)	F1 Score (%)	Training Accuracy (%)	Testing Accuracy (%)
**Original CKHK-22 Datasets**	**ADaDR-22**	89.92	93.47	86.76	97.74	93.47
	**ADaR-22**	92.12	91.58	85.61	96.95	91.58
	**DaRD-22**	95.31	94.96	93.47	98.64	95.96
**Datasets**	**Ensemble Fusion CNNs**	**Precision (%)**	**Recall (%)**	**F1 Score (%)**	**Training Accuracy (%)**	**Testing Accuracy (%)**
**Greyscale CKHK-22 Datasets**	**ADaDR-22**	89.23	87.89	86.42	92.11	87.89
	**ADaR-22**	84.92	83.09	82.57	85.54	83.09
	**DaRD-22**	90.46	88.79	87.62	95.56	88.79
**Datasets**	**Ensemble Fusion CNNs**	**Precision (%)**	**Recall (%)**	**F1 Score (%)**	**Training Accuracy (%)**	**Testing Accuracy (%)**
**LBP CKHK-22 Datasets**	**ADaDR-22**	70.67	69.92	67.41	72.56	69.92
	**ADaR-22**	71.11	68.85	67.96	71.87	68.85
	**DaRD-22**	75.11	73.54	72.67	75.89	73.54
**Datasets**	**Ensemble Fusion CNNs**	**Precision (%)**	**Recall (%)**	**F1 Score (%)**	**Training Accuracy (%)**	**Testing Accuracy (%)**
**Feature Fusion CKHK-22 Datasets**	**ADaDR-22**	95.69	94.96	93.37	95.67	94.96
	**ADaR-22**	94.98	93.77	92.59	95.08	93.77
	**DaRD-22**	96.98	97.12	95.98	98.72	97.89

**Table 6 bioengineering-10-00738-t006:** Performance metric for ensemble fusion CNN-DarD-22 using the feature fusion CKHK-22 mixed dataset.

Classes	Precision	Recall	F1 Score	Support
bbps-0-1	0.99	0.98	0.99	594
bbps-2-3	0.99	0.99	0.99	1035
cecum	0.88	0.99	0.93	1809
dyed-lifted-polyps	0.58	0.92	0.71	1803
dyed-resection-margins	0.86	0.43	0.57	1791
Non-Polyps	0.98	0.96	0.97	771
polyps	0.96	0.82	0.88	2604
pylorus	0.97	1	0.98	1800
retroflex-stomach	0.99	0.99	0.99	690
z-line	1	0.99	0.99	1740

**Table 7 bioengineering-10-00738-t007:** Examining colorectal cancer detection procedures: a 2021–2022 comparative study.

Author	Method	Model Approach	Dataset	Time Elapsed	Accuracy (%)
**Omneya Attallah et al. (2021)** [17]	**GastroCADx**	AlexNet, DarkNet19, ResNet50 and DenseNet-201, DWT and DCT functions, SVM	Kvasir2,	3 h	97.3
			Hyper Kvasir	3 h	99.7
**Liew et al. (2021)** [11]	**Ensemble classifier**	ResNet50 + Adaboost, AlexNet, GoogLeNet, and VGG-19	CVC Clinic DB	2.5 h	97.91
**Pallabi Sharma et al. (2022)** [49]	**Ensemble classifier**	ResNet101, GoogleNet and XceptionNet	CVC Clinic DB, Aichi Medical Dataset	2.45 h	98.3
			Kvasir2,	2.25 h	97
**Nisha J.S et al. (2022)** [50]	**DP-CNN**	Dual Path CNN	CVC Clinic DB, ETIS-Larib	2 h	99.6
**Maryem Souaidi et al. (2022)** [21]	**MP-FSSD**	VGG16 with feature Fusion Module	CVC Clinic DB, WCE dataset	2.5 h	91.56
**Akella S Narasimha Raju et al. (2022)** [51]	**ColoRectalCADx**	ResNet-50V2, DenseNet-201, VGG16, LSTM and SVM	Hyper Kvasir Balanced	2.15 h	98.91
			Mixed Dataset Balanced	2.10 h	96.13
**Proposed Model (2023)**	**EnsemDeepCADx**	Ensemble CNN DaRD-22, BLSTM, SVM with feature fusion	CKHK-22 Mixed Dataset	2 h	97.89

## Data Availability

Collected data is from publicly accessible colonoscopy datasets, CVC clinic DB, Kvasir 2, and Hyper Kvasir for the project. Even though this work with internal human organs is available to the public via internet sources, it has not been deemed ethical by official authorities. Data are publicly available from the following websites: CVC clinic DB Dataset was obtained from https://www.kaggle.com/datasets/balraj98/cvcclinicdb (accessed on 3 May 2023); Kvasir2 Dataset was obtained from https://datasets.simula.no/kvasir/ (accessed on 3 May 2023); Hyper Kvasir Dataset was obtained from https://datasets.simula.no/hyper-kvasir/ (accessed on 3 May 2023). In this research paper, the combining of three datasets is presented as a new dataset known as a mixed dataset.

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
