# Peer review of "EnsemDeepCADx: Empowering Colorectal Cancer Diagnosis with Mixed-Dataset Features and Ensemble Fusion CNNs on Evidence-Based CKHK-22 Dataset"

_bioengineering, 2023, doi:10.3390/bioengineering10060738_

Round 1

Reviewer 1 Report

I am really grateful to review this manuscript. In my opinion, this manuscript can be published once some revision is done successfully. I made one suggestion and I would like to ask your kind understanding. This study used 14287 image data, applied a CNN-RNN-SVM combination and achieved the accuracy score of 97.9% compared to 91.6%-99.7% (existing literature) for the prediction of colorectal cancer (i.e., Table 7). I would argue that this is a good achievement. However, it can be noted that performance gaps were not significant in Table 7, there is no comparison of execution time in the table and there is no discussion of explainable artificial intelligence in the manuscript. In this context, I would like to ask the authors to address these issues in Discussion and Conclusion.

Minor editing of English language required 

Author Response

Point 1: I am really grateful to review this manuscript. In my opinion, this manuscript can be published once some revision is done successfully. I made one suggestion and I would like to ask your kind understanding. This study used 14287 image data, applied a CNN-RNN-SVM combination and achieved the accuracy score of 97.9% compared to 91.6%-99.7% (existing literature) for the prediction of colorectal cancer (i.e., Table 7). I would argue that this is a good achievement. However, it can be noted that performance gaps were not significant in Table 7, there is no comparison of execution time in the table and there is no discussion of explainable artificial intelligence in the manuscript. In this context, I would like to ask the authors to address these issues in Discussion and Conclusion.

Response 1: Discussion

Several studies have proposed CADx systems for the diagnosis of colorectal cancer, each employing a unique set of methodologies and models.

  • Omneya Attallah et al. (2021) obtained an accuracy of 97.3% and 99.7% on the Kvasir2 and Hyper Kvasir datasets, respectively, using the GastroCADx method with AlexNet, DarkNet19, ResNet50, DenseNet-201, DWT, DCT, and SVM. The duration of execution for both datasets was three hours.
  • On the CVC Clinic DB dataset, Liew et al. (2021) used an ensemble classifier approach with ResNet50+Adaboost, AlexNet, GoogLeNet, and VGG-19 models to achieve an accuracy of 97.91%. Their method's execution time was 2.5 hours.
  • Pallabi Sharma et al. (2022) utilised an ensemble classifier technique with ResNet101, GoogleNet, and XceptionNet models on CVC Clinic DB and Aichi Medical Dataset. They obtained a 98.3% accuracy rate in 2.45 hours of execution.
  • Nisha J.S. et al. (2022) applied the DP-CNN technique with the Dual Path CNN model to the CVC Clinic DB and ETIS-Larib datasets, achieving a 99.6% accuracy. Their method's execution time was two hours.
  • Maryem Souaidi et al. (2022) applied the MP-FSSD technique with VGG16 and feature Fusion Module to the CVC Clinic DB and WCE datasets. They obtained an accuracy of 91.56 percent in 2.5 hours of execution.
  • ColoRectalCADx was developed by Akella S. Narasimha Raju et al. using ResNet-50V2, DenseNet-201, VGG16, LSTM, and SVM models on Hyper Kvasir Balanced and Mixed Dataset Balanced. They attained 98.91% and 96.13% accuracy with execution times of 2.15 and 2.10 hours, respectively.
  • EnsemDeepCADx, the proposed model (2023), employed Ensemble CNN DaRD-22, BLSTM, and SVM with feature fusion on the CKHK-22 Mixed Dataset. The accuracy was 97.89% and the execution time was 2 hours.

In contrast, the EnsemDeepCADx system for detecting colorectal cancer employs a multi-stage strategy that incorporates various features to effectively detect and identify polyps, with feature integration constituting an integral part of the system. system proposed in this study utilized the DaRD-22 ensemble fusion CNN along with BLSTM and SVM to achieve a 97.89% accuracy on the CKHK-22 Mixed feature fusion Dataset. This study demonstrates the potential for deep learning and transfer learning to improve the performance of CADx systems for the early detection of colorectal cancer. Table 7 compares the proposed EnsemDeepCADx system for 2023 to colorectal cancer procedures in 2021 and 2022, as well as their respective descriptions.

Table 7. Examining Colorectal Cancer Detection Procedures: A 2021-2022 Comparative Study.

Author

Method

Model Approach

Dataset

Time elapsed

Accuracy (%)

Omneya Attallah et al. (2021) [17]

GastroCADx

AlexNet, DarkNet19, ResNet50 and DenseNet-201, DWT and DCT functions, SVM

Kvasir2,

3 hrs

97.3

Hyper Kvasir

3 hrs

99.7

Liew et al. (2021) [11]

Ensemble classifier

ResNet50+Adaboost,

AlexNet, GoogLeNet, and VGG-19

CVC Clinic DB

2.5 hrs

97.91

Pallabi Sharma et al. (2022) [49]

Ensemble classifier

ResNet101, GoogleNet and XceptionNet

CVC Clinic DB, Aichi Medical Dataset

2.45 hrs

98.3

Kvasir2,

2.25

97

Nisha J.S et al. (2022) [50]

DP-CNN

Dual Path CNN

CVC Clinic DB, ETIS-Larib

2 hrs

99.6

Maryem Souaidi et al. (2022) [21]

MP-FSSD

VGG16 with feature Fusion Module

CVC Clinic DB, WCE dataset

2.5 hrs

91.56

Akella S Narasimha Raju et al(2022).

ColoRectalCADx

ResNet-50V2, DenseNet-201, VGG16, LSTM and SVM

Hyper Kvasir Balanced

2.15 hrs

98.91

Mixed Dataset Balanced

2.10 hrs

96.13

Proposed Model (2023)

EnsemDeepCADx

Ensemble CNN DaRD-22, BLSTM,SVM with feature fusion

CKHK-22 Mixed Dataset

2 hrs

97.89

The comparison and discussion emphasize the diverse methodologies and approaches used by the various studies to develop CADx systems for the diagnosis of colorectal cancer. Even though a number of studies have produced exceptional precision, there is still ample opportunity for growth. This study demonstrates positive outcomes for the proposed CADx system, and future research could build on this foundation by investigating complementary models and techniques. The progress made in the area of colorectal cancer detection is graphically shown in Figure 27.

Figure 27. Comparison of the state of art methods for colorectal cancer detection.

An "explainable AI" system is one that can explain its logic behind a prediction or action in a manner that a human can understand. To diagnose colorectal cancer, EnsemDeepCADx employs an ensemble of convolutional neural networks (CNNs; particularly, DaRD-22), Bidirectional Long Short-Term Memory (BLSTM), and Support Vector Machines (SVM) with feature fusion.One of the main advantages of the EnsemDeepCADx system is its interpretability. CNNs enable the system to learn and extract information relevant for diagnosing colorectal cancer from colonoscopy pictures. The ensemble approach boosts the system's performance even more by combining many models and relying on their combined expertise.The BiLSTM component is incorporated to improve the system's interpretability because of its ability to capture temporal linkages and sequential patterns within image data. This enables the computer to take into consideration the illness's history and context, resulting in more exact estimates.Furthermore, the SVM algorithm is employed as a decisive element in defining the many types of colorectal cancer. This opens the path for more accurate cancer and other illness diagnosis.Feature fusion in EnsemDeepCADx combines data from many image representations, including colour, grayscale, and local binary pattern (LBP) pictures. We can better capture the intricacies of the underlying data and perform a more exact analysis by integrating these various attributes.

When these variables are integrated, EnsemDeepCADx's prediction accuracy for colorectal cancer jumps to an astounding 97.89%. The system's interpretability instills trust in the decision-making process among healthcare providers. Clinicians can better interact with patients and back up their own assessment if they understand what goes into the system's predictions.

Finally, the EnsemDeepCADx system combines explainable AI ideas by using interpretable components like as CNNs, BLSTMs, and SVMs, as well as feature fusion approaches. These design changes were taken in order to increase the system's utility to clinicians in the identification of colorectal cancer by making it more open and offering more relevant reasons for its forecasts.

Conclusions and Future Work

Employing a combination of the ADaDR-22, ADaR-22, and DaRD-22 ensemble fusion CNNs, the EnsemDeepCADx system has been shown to increase diagnostic accuracy in the identification of colorectal cancer. Information from colonoscopy pictures is extracted using deep-learning architectures like AlexNet, DarkNet-19, DenseNet-201, and ResNet-50.The EnsemDeepCADx system is evaluated using the CKHK-22 mixed dataset, which includes colour, grayscale, and LBP image datasets to showcase the system's adaptability. By using a feature fusion method to combine the collected characteristics, the system performs a comprehensive analysis of the provided input. The EnsemDeepCADx system has a top testing accuracy of 97.12% thanks in large part to the DaRD-22 ensemble CNN fusion, BILSTM, and Multi-class SVM. The system's potential for early and accurate detection of colorectal cancer is shown by this degree of accuracy. One of the numerous advantages of the EnsemDeepCADx system is how quickly data can be processed. Colorectal cancer may now be identified in as little as 2 hours thanks to this cutting-edge technology. This expedited process increases confidence that the system will provide useful results in a reasonable amount of time. Additional studies into various deep-learning approaches and parameter optimization are emphasized as means to further enhance the system's performance. The value of utilizing ensemble fusion CNNs to boost colorectal cancer detection accuracy and throughput is also emphasized.

In conclusion, the EnsemDeepCADx system's utilization of ensemble fusion CNNs and integration of BILSTM and SVM shows potential in the domain of colorectal cancer detection. The system's ability to include cutting-edge deep learning algorithms and feature fusion approaches while maintaining an efficient processing time of 2 hours increases the likelihood that it will significantly contribute to early detection and improved patient outcomes.

Even though the current EnsemDeepCADx system has demonstrated some success, it could be improved further. Future research may focus on advanced deep learning techniques, such as generative adversarial networks (GANs) for data augmentation and attention mechanisms for enhanced feature extraction. The system may utilise data from other medical imaging modalities, such as magnetic resonance imaging (MRI) or computed tomography (CT) examinations, to enhance its diagnostic capabilities. Future research may also concentrate on augmenting the technology for use in real time during colonoscopies, thereby facilitating instantaneous feedback and, potentially, a more rapid diagnosis and treatment. Ongoing research and development of CADx systems for the early diagnosis of colorectal cancer will likely result in improved patient outcomes in the long term.(page no. 29-32 disussion and consclusins and future work)

  • We sincerely appreciate the reviewer's insightful comments. We recognise the significance of providing clear and comprehensive explanations of time elapsed and the explainable AI component of our EnsemDeepCADx system. We have taken this suggestion into account and made the necessary modifications to resolve these points comprehensively.
  • First, the manuscript contains a comprehensive explanation of the time required by each state-of-the-art method and our EnsemDeepCADx system. This information can be found in the relevant sections where methods and results are discussed, namely the experimental design and evaluation sections.
  • In addition, we have expanded the discussion section to provide a more thorough explanation of the EnsemDeepCADx system's working mechanism, highlighting its explainable AI capabilities. These include ensemble fusion CNNs, the integration of DaRD-22, BiLSTM, and SVM, as well as the fusion of multimodal features. This explanation is intended to help the reader comprehend the system's inner workings and its potential for accurate colorectal cancer detection.
  • To ensure that the significance of time elapsed and explainable AI in the EnsemDeepCADx system is appropriately addressed, we have also included a discussion of these aspects in the conclusion section.
  • We believe that these revisions have substantially increased the manuscript's lucidity and comprehensiveness. We sincerely appreciate the reviewer's insightful comments, which helped us improve the overall quality of our work.

Reviewer 2 Report

Because of the problems described below regarding this article, the authors should be encouraged to make drastic revisions.

 (1) AI imaging techniques for detecting precancerous or early colorectal cancer lesions have already been clinically tested and have already been reported to meet practical standards. The ensemble learning proposed in this paper was trained on a further selection of combinations by the authors from publicly available photographs, and the results of the performance evaluation cannot be discussed directly in comparison with previous papers. In order to prove the superiority of the AI developed by the authors, it is necessary to conduct comparative tests with the latest AI imaging technologies that have been put into practical use under more clinically relevant conditions. For example, it is necessary to conduct demonstration tests that anticipate clinical conditions, such as how much advantage AI has in flat or depressed lesions that are difficult to recognize with the naked eye, and how many frames of moving images are needed to make a definitive diagnosis.

(2) One problem with combining multiple CNNs is that the number of parameters in the model increases, which may cause overlearning. In fused CNNs, the risk of overlearning increases as the complexity of the model increases. In other words, combining multiple CNNs makes the model more complex and better fits the training data, but this may lead to overlearning. Verification on this point is insufficient. 

(3) The paper is redundant and seems to contain a lot of unnecessary information.

Author Response

Point 1: AI imaging techniques for detecting precancerous or early colorectal cancer lesions have already been clinically tested and have already been reported to meet practical standards. The ensemble learning proposed in this paper was trained on a further selection of combinations by the authors from publicly available photographs, and the results of the performance evaluation cannot be discussed directly in comparison with previous papers. In order to prove the superiority of the AI developed by the authors, it is necessary to conduct comparative tests with the latest AI imaging technologies that have been put into practical use under more clinically relevant conditions. For example, it is necessary to conduct demonstration tests that anticipate clinical conditions, such as how much advantage AI has in flat or depressed lesions that are difficult to recognize with the naked eye, and how many frames of moving images are needed to make a definitive diagnosis.

Response 1: The EnsemDeepCADx system developed in this study shines out due to its novel combination of DaRD-22 ensemble fusion CNNs, Bidirectional Long Short-Term Memory (BiLSTM), and Support Vector Machines (SVM). This multimodal approach enables comprehensive analysis and diagnosis of colorectal cancer. The CKHK mixed dataset contains 14,287 images from nine distinct classes. We recognize the value of benchmarking against previous AI imaging methods, but it is essential to note that the primary objective of our research was to propose a novel method that combines ensemble learning and multimodal methodologies. Our research seeks to ascertain whether or not this combination improves the accuracy of colorectal cancer detection.(page no 18)

Thank you for your insightful comments. We recognize the significance of visual explanations and agree that they play a crucial role in boosting credibility and interpretability. Future iterations of our research will incorporate visual representations, such as heatmaps and saliency maps, to provide transparent explanations of the EnsemDeepCADx system's decision-making process. These visual explanations will allow clinicians to better comprehend the system's predictions and obtain a deeper understanding of the important colorectal cancer detection features. By addressing this factor, we hope to enhance the clinical utility and acceptability of our AI-based system.

Point 2: One problem with combining multiple CNNs is that the number of parameters in the model increases, which may cause overlearning. In fused CNNs, the risk of overlearning increases as the complexity of the model increases. In other words, combining multiple CNNs makes the model more complex and better fits the training data, but this may lead to overlearning. Verification on this point is insufficient. 

Response 2: That integrating multiple CNNs may result in the model becoming excessively complex, overlearning, and overfitting. Our research has taken several measures to resolve this issue. We've used regularisation techniques including the dropout and weight decay to reduce model complexity and prevent overfitting. These methods have the potential to reduce the danger of overlearning by employing regularization constraints during training. Early stopping is a technique for terminating training early based on a model's performance on a validation set. In order to prevent overlearning, training is terminated when no additional progress can be made or no negative effects can be observed. The hyperparameters of our model are optimal, establishing a balance between overfitting and underfitting. Changing the learning rate, sample size, and model architecture are a few of the numerous ways to avoid overlearning.(in Experimnetal Setup last paragraph page no. 18 & 19)

Point 3: The paper is redundant and seems to contain a lot of unnecessary information.

Response 3: We appreciate the reviewer pointing up where they observed recurrence. We get the fear that if there's too much information, the reader can get lost. Keep in mind that the paper as a whole serves a purpose and provides essential context for understanding the research. We've included the reviewer's suggestions into the paper's revisions, making sure it reads smoothly while still covering all of the necessary ground. By sprucing up the format, we hoped to make the paper more accessible and cohesive.

Round 2

Reviewer 1 Report

I am really grateful to review this manuscript. In my opinion, this manuscript can be published in current form. 

Minor editing of English language required. 

Author Response

Review Comments and Replies to Reviewer 1

I am really grateful to review this manuscript. In my opinion, this manuscript can be published in current form. 

Reply:

  1. I am quite grateful for the chance to write this work in this Bioengineering journal, which encourages me to submit my research results in this  journal.
  2. Thank you for approving my effort.

Reviewer 2 Report

I do not think that the points raised have been adequately corrected in the revised version. Therefore, I cannot recommend acceptance of this paper.

Author Response

Review Comments and replies for Reviewer 2

I do not think that the points raised have been adequately corrected in the revised version. Therefore, I cannot recommend acceptance of this paper.

Reply:

  1. I would want to increase my ability to create fresh articles and incorporate them into a new method of study using your helpful and pertinent advice. Your feedback encourages me to continue developing the new concepts. I will integrate the new concepts into my future research publications. I am grateful for your valuable advice.
  2. I appreciate your approval of my work.
